# Length regulation of multiple flagella that self-assemble from a shared pool of components

**Thomas G Fai[1], Lishibanya Mohapatra[2], Prathitha Kar[3], Jane Kondev[2], Ariel Amir[3]\***

[1]Department of Mathematics, Brandeis University, Waltham, United States; [2]Department of Physics, Brandeis University, Waltham, United States; [3]Paulson School of Engineering and Applied Sciences, Harvard University, Cambridge, United States

**Abstract** The single-celled green algae *Chlamydomonas reinhardtii* with its two flagella—microtubule-based structures of equal and constant lengths—is the canonical model organism for studying size control of organelles. Experiments have identified motor-driven transport of tubulin to the flagella tips as a key component of their length control. Here we consider a class of models whose key assumption is that proteins responsible for the intraflagellar transport (IFT) of tubulin are present in limiting amounts. We show that the limiting-pool assumption is insufficient to describe the results of severing experiments, in which a flagellum is regenerated after it has been severed. Next, we consider an extension of the limiting-pool model that incorporates proteins that depolymerize microtubules. We show that this 'active disassembly' model of flagellar length control explains in quantitative detail the results of severing experiments and use it to make predictions that can be tested in experiments.

**\*For correspondence:**
arielamir@seas.harvard.edu

**Competing interests:** The authors declare that no competing interests exist.

## Introduction

The size regulation of cellular organelles is a fundamental problem in biology (*Marshall, 2016*; *Milo and Phillips, 2015*). For example, nuclear size is tightly coupled with cell size across a wide range of species (*Hara and Merten, 2015*) and the loss of this coupling mechanism is implicated in various types of cancer (*Zink et al., 2004*).

A striking example of organelle size control in eukaryotes is the single-celled algae *Chlamydomonas reinhardtii* (*Figure 1*), which uses two flagella to move through its aqueous environment. The backbone of each flagellum is an assembly known as the *axoneme* that consists of nine microtubule doublets arranged in a ring around a central pair of microtubules (*Fawcett and Porter, 1954*; *Witman et al., 1972*). Unlike the dynamic instability of cytoplasmic microtubules, which can alternate between rapidly shortening 'catastrophe' and stable 'rescue' states depending on whether or not the tip is bound to GTP, microtubules in the axoneme exist in a highly stable state (*Behnke and Forer, 1967*; *Orbach and Howard, 2019*). This stability reflects a tight control over flagellar lengths, the loss of which has dramatic physiological consequences; mutants with longer flagella have decreased swimming velocities and beat frequencies (*Khona et al., 2013*) compared to wild type cells and mutants with unequal flagellar lengths are observed to spin around in circles (*Tam et al., 2003*).

A key process contributing to the assembly of flagella is the continual transport of proteins from the flagellar base to tip and back. The original evidence for this intraflagellar transport (IFT) was provided around 25 years ago by experimental observations in *Chlamydomonas* of particles moving processively along the flagellum at constant speed (*Kozminski et al., 1993*). In the time since, a

**Figure 1.** Experimental background. (a) Electron microscopy images of the biflagellate green algae *Chlamydomonas* and its flagella captured by Elisa Vannuccini and Pietro Lupetti (University of Siena, Italy) and reproduced from *Morga and Bastin (2013)* under the Creative Commons Attribution License CC BY 2.0 [http://creativecommons.org/licenses/by/2.0]). The inset shows the whole organism (scale bar 5 µm) and the close-up shows the flagellar basal body (BB), transition zone (TZ), and cell wall (CW) (scale bar 1 µm). (b) Severing experiments: after one flagellum is severed, the two flagella equalize at a shorter length and then grow together to the original steady-state length. This is demonstrated in experimental data of 20 severing experiments from *Ludington et al. (2012)* provided by the authors. The green and blue shaded regions show the mean plus or minus one standard deviation.

significant body of work has revealed the many proteins and biochemical pathways that coordinate this complex process in *Chlamydomonas* and other organisms such as *C. elegans*, as described in several review articles (*Prevo et al., 2017*; *Scholey, 2003*; *Scholey, 2008*; *Cole, 2003*; *Rosenbaum et al., 1999*; *Rosenbaum and Witman, 2002*; *Rosenbaum et al., 1999*).

Already at the time of its discovery, IFT was hypothesized to play a role in flagellar length control by transporting building blocks to the tip of the flagellum (*Kozminski et al., 1993*). IFT particles containing tubulin are transported along the flagellum by two different motor proteins: kinesin-2 transports IFT particles from the flagellar base to tip (the *anterograde* direction) whereas dynein carries IFT particles from the tip to the base (the *retrograde* direction). As shown in subsequent work (*Song and Dentler, 2001*; *Marshall and Rosenbaum, 2001*; *Buisson et al., 2013*), these flagellar proteins are continually exchanged with those localized to the basal body of the flagellum, represented schematically in *Figure 2a* as a *basal pool*.

These observations motivated the development of mathematical models of flagellar length dynamics such as the balance point model (*Marshall and Rosenbaum, 2001*; *Marshall et al., 2005*). In the balance point model, there is a continual competition between assembly and disassembly, and either the rate of assembly, the rate of disassembly, or both, may be length-dependent. The steady-state length is determined by the point at which the assembly and disassembly processes come into balance.

Up to now, the balance point model has been considered primarily for a single flagellum and the focus has been on assembly being the length dependent process that leads to length control

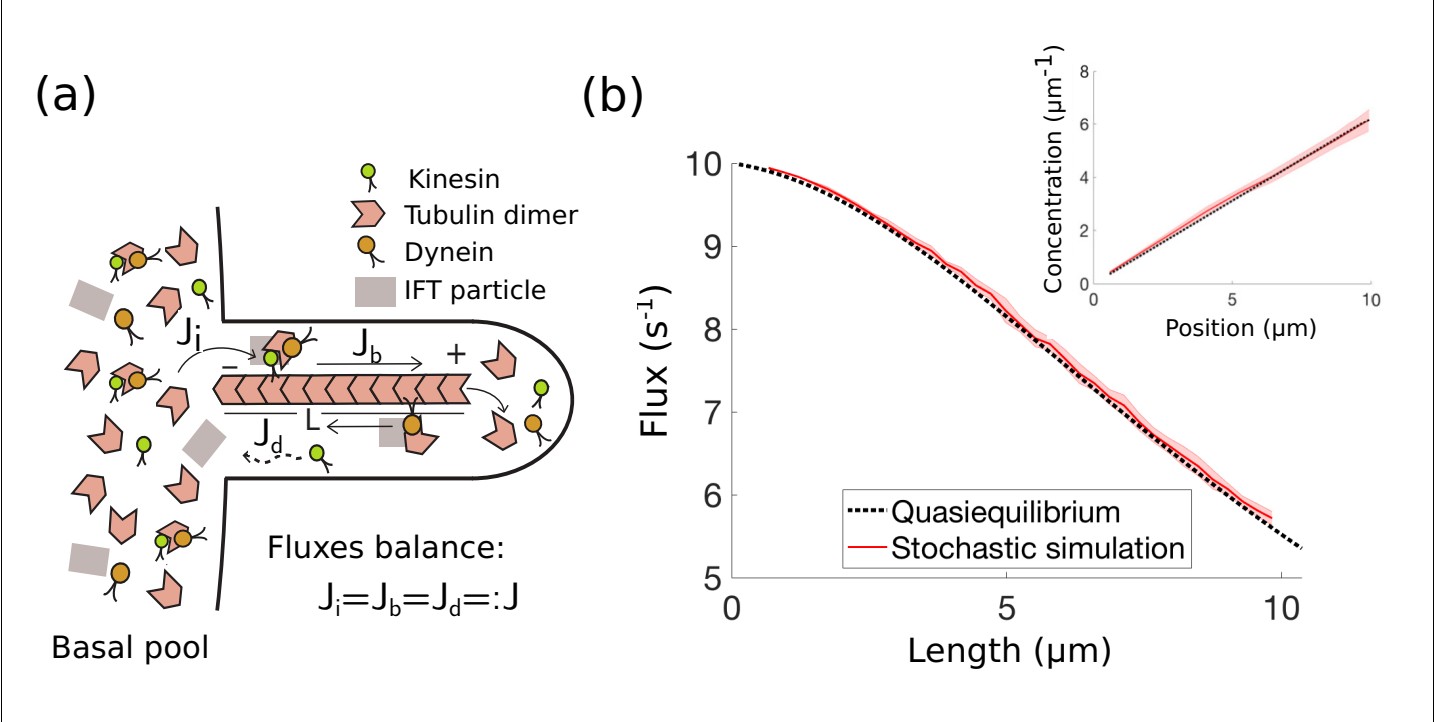

**Figure 2.** IFT in *Chlamydomonas* with diffusive return of kinesin-2 to the base. (a) Kinesin-2, dynein, and tubulin combine to form a complex with other IFT components in the basal pool and are injected into the flagellum. The kinesin-2 motors move toward the tip of the flagellum, where the complex eventually breaks down into its constituent parts. Dynein motors move in the retrograde direction carrying back some, but not all IFT components. Specifically, kinesin motors are not part of the retrograde IFT complex and they diffuse back to the base. In steady-state, the injection flux $J_i$, ballistic flux $J_b$, and diffusive flux $J_d$ are all equal. (b) IFT flux vs. length, comparison between the quasi-steady state approximation (*Equation 4*) and stochastic simulations (Appendix 1). The shaded red area represents the mean and standard deviation computed from 10 realizations of the stochastic simulations. (inset) Kinesin-2 concentration vs. position along the flagellum (steady-state approximation (*Equation 31*)).

(*Marshall and Rosenbaum, 2001*). However, other formulations of the balance point model, such as the case in which the disassembly rate is also length-dependent, have received considerably less attention. In this work, we revisit the balance point model and particularly the assumption of a constant rate of disassembly in the context of simultaneous length control of multiple flagella assembled from a shared pool of biomolecules.

We limit our theoretical exploration to the space of models defined by the following processes: IFT particle assembly and injection at the flagellar base, motion of IFT proteins along the flagellum, and tubulin polymerization and depolymerization at the flagellar tip (see schematic *Figure 2a*). We further assume that IFT particle injection satisfies first-order chemical kinetics and allow for a control mechanism that regulates protein levels in the basal pool. Note that this model space does not include all possibilities. In particular, it does not include the time-of-flight model (*Wren et al., 2013*), in which additional reactions affect protein state inside the flagellum (e.g. proteins enter in an activated state and deactivate at some rate).

Nevertheless, our work retains a high degree of generality. For example, it allows for different modes of coupling between proteins in basal pools (shared or separate), different modes of IFT motion (ballistic or diffusive and processive or non-processive), and general depolymerizing activity. Within the model space outlined, our main results hold independent of these details. Notably, we find that length-independent disassembly of microtubules cannot account for the experimental results, whereas incorporating length-dependent disassembly (e.g. through the ballistic-to-diffusive motion of a depolymerizing protein) leads to reasonable agreement with the experiments.

We analyze these models at two levels of detail, both using detailed agent-based stochastic simulations and a reduced description in terms of ordinary differential equations (ODEs) and compare the results of simulations to experimental data. It is known from experiments on *Chlamydomonas*

that (i) its two flagella reach steady-state lengths of about 10 μm, and that (ii) the flagellar lengths are correlated since, if one flagellum is severed, the remaining flagellum shortens until it reaches the length of the growing, previously severed flagellum (*Rosenbaum et al., 1969*). This latter protocol is typically referred to as a 'severing' or 'long-zero' experiment.

As shown in *Figure 1b*, the severing experiments contain two separate timescales. There is an initial fast timescale of 10–20 min over which the flagellar lengths first equalize, followed by a slower timescale of 60–90 min over which the lengths increase simultaneously to their original lengths. With respect to the short time behavior, we find that none of the candidate models assuming constant disassembly capture the rapid length equalization observed experimentally. Moreover, if an external control mechanism is added that replenishes proteins lost due to severing over a slower timescale, the constant disassembly models within our model space fail to control lengths at all.

Motivated by experimental observations of microtubule-depolymerizing proteins within the flagellum (*Piao et al., 2009*; *Luo et al., 2011*; *Hilton et al., 2013*), we subsequently consider a model in which the disassembly rate is dependent on the local concentration of a depolymerizing protein at the flagellar tips. This gives rise to a length-dependent disassembly rate, which is a departure from existing models that assume constant disassembly. We find that this model is consistent with the rapid length equalization observed in experiments. Further, upon adding a control mechanism on protein levels in the basal protein pool, the model is able to capture the slow return to the original steady-state lengths observed experimentally. In the Discussion, we examine this model in light of the currently available experimental data and discuss possible candidates for the depolymerizing protein.

## Results

### Limiting motor pool gives rise to length-dependent injection rate

It is known from experiments that the injection rate of IFT particles in *Chlamydomonas* is length-dependent (*Dentler, 2005*). In this section, we show how length-dependent injection is a consequence of mass action kinetics of proteins available in limiting amounts.

We first consider the flagellar length dynamics of a single flagellum. Our approach initially closely follows *Hendel et al. (2018)*, although important differences will appear later on in the case of two flagella. The key biochemical variables are the tubulin dimers that make up the flagellar axoneme and the molecular motors kinesin-2 and dynein that transport IFT particles from the base to the tip and back. IFT particles combine in the basal pool with kinesin-2, tubulin and dynein to form a complex that is injected into the flagellum (*Cole et al., 1998*); see the schematic in *Figure 2a*.

Here we consider the case that the rate-limiting molecule is kinesin-2, although similar results would apply to any other protein being rate-limiting for IFT assembly. Denoting the number of free molecular motors in the pool at the flagellar base by $M_f$, the injection flux $J_i$ of IFT particles into the flagellum satisfies

$$J_i = k_{\mathrm{on}} M_f. \tag{1}$$

Note that we consider the cell volume to be fixed, in which case biomolecule numbers and concentrations may be used interchangeably.

First we consider the case in which the total number of motors is conserved. Later on we will also consider the case in which motor concentrations in basal pools are regulated by an external control mechanism, motivated by the severing experiments described previously in which length recovery indicates replenishment of protein levels.

Kinesin-2 has been shown to undergo ballistic transport in the anterograde direction and diffusive motion in the retrograde direction (*Chien et al., 2017*). The total number of motors $M$ satisfies $M = M_f + M_b + M_d$, where $M_b$ is the number of motors moving ballistically on the flagellum in IFT particles and $M_d$ is the number of motors moving diffusively. Therefore we may rewrite *Equation 1* as

$$J = k_{\mathrm{on}}(M - M_b - M_d). \tag{2}$$

The intraflagellar dynamics are fast compared to changes in length. The timescale of flagellar length dynamics, for example the recovery time after severing, is of the order of 10 min, whereas the

molecular motors involved in IFT take at most tens of seconds to traverse the length of the flagellum by either ballistic or diffusive motion. Based on this separation of timescales, we treat IFT as a quasi-steady state process in which the injection flux $J_i$, diffusive flux $J_b$, and ballistic flux $J_d$ are balanced, that is

$$J_i = J_b = J_d =: J. \qquad (3)$$

As shown in Materials and methods, the quasi-steady state assumption of flux balance together with mass action kinetics of injection expressed in *Equation 1* imply that

$$J = \frac{k_{\mathrm{on}}M}{1 + k_{\mathrm{on}}L/v + k_{\mathrm{on}}L^2/2D}, \qquad (4)$$

where $k_{\mathrm{on}}$ is the rate constant of motor injection, $M$ is the total number of motors, $v$ is the ballistic motor speed in IFT, and $D$ is the diffusion coefficient of motors in the flagellum.

We have validated the quasi-steady state assumption used to derive *Equation 4* by comparing the IFT particle flux and the concentration of diffusing motors in the flagellum to the results of stochastic simulations (see Appendix 1), in which the dynamics of motors as well as the dynamics of microtubule assembly are taken into account explicitly; see *Figure 2b*. Parameter values used in simulations are provided in *Table 1*. (Note that in *Equation 4* and throughout the manuscript, terms written in the form $x/yz$ are shorthand for $x/(yz)$, for example $k_{\mathrm{on}}L^2/2D$ is to be read as $k_{\mathrm{on}}L^2/(2D)$.)

IFT particle injection arising from a finite number of motors shared between the flagellum and the basal pool therefore leads to a length-dependent flux. This result holds true regardless of the identity of the rate-limiting IFT protein. However, the scaling of formula *Equation 4* with length depends on whether ballistic or diffusive transport dominates. In the limit $D \gg Lv$, the $1/L$ scaling of *Marshall and Rosenbaum (2001)* is recovered whereas in the limit $D \ll Lv$, we recover the $1/L^2$ scaling of *Hendel et al. (2018)*. Note that a distinction between our model and these previous works is the presence of a constant term in the denominator of *Equation 4*, which implies that in our formulation the flux does not blow up at $L = 0$.

**Table 1.** Parameter values and definitions.

| Symbol | Definition | Value | Units | References |
|---|---|---|---|---|
| **Parameters** | | | | |
| $L_{ss}$ | Steady-state length | 10–12 | μm | *Marshall and Rosenbaum, 2001*; *Rosenbaum et al., 1969* |
| $T/N$ | Tubulin pool per flagellum | 38–47 | μm | *Marshall et al., 2005* |
| $d$ | Disassembly speed | 0.5 | μm/min | *Marshall and Rosenbaum, 2001*; *Ludington et al., 2012* |
| $v$ | IFT speed | 2.5–3 | μm/s | *Kozminski et al., 1993*; *Buisson et al., 2013* |
| $D$ | Diffusion coefficient | 1.7 | μm$^2$/s | *Chien et al., 2017* |
| $\gamma k_{\mathrm{on}}M/N$ | Assembly rate per tubulin | $2.3 \times 10^{-2}$ –$3.6 \times 10^{-2}$ | min$^{-1}$ | Fit |
| $k_{\mathrm{on}}$ | Injection rate constant | 0.8–4 | min$^{-1}$ | Fit |
| $\gamma$ | Prefactor in *Equation 5* | $2.5 \times 10^{-4}$ | | Estimate (Appendix 2) |
| **Variables** | | | | |
| $N$ | Number of flagella | | | |
| $T_f$ | Free tubulin | | μm | |
| $M$ | Total motors | | | |
| $M_f$ | Free motors | | | |
| $M_b$ | Ballistic motors | | | |
| $M_d$ | Diffusing motors | | | |
| $J$ | Flux | | min$^{-1}$ | |
| $c_d(x)$ | Motor concentration | | μm$^{-1}$ | |
| $\overline{c_d}$ | Average concentration | | μm$^{-1}$ | |

In our model the assembly rate is determined by the rate of tubulin transport to the flagellar tip, as in *Marshall and Rosenbaum (2001)* and *Hendel et al. (2018)*. Given mass-action kinetics of IFT particle assembly in the basal pool, this is simply the flux $J$ of IFT particles times the amount of free tubulin $T_f$, so that the growth rate is given by the following ODE:

$$\frac{\mathrm{d}L}{\mathrm{d}t} = \gamma J T_f - d, \tag{5}$$

where $d$ is the disassembly speed (assumed constant for now) and $\gamma$ is a constant. The total amount of tubulin $T$ is assumed to be conserved for now so that $T = T_f + L$, where tubulin is measured in units of corresponding flagellar length. As previously mentioned, later on we will consider the case in which protein levels in basal pools are not conserved and are instead monitored and regulated by an external control mechanism.

Substituting the expression *Equation 4* for the flux into the above growth rate results in

$$\frac{\mathrm{d}L}{\mathrm{d}t} = \frac{\gamma k_{\mathrm{on}} M}{1 + k_{\mathrm{on}} L/v + k_{\mathrm{on}} L^2/2D}(T - L) - d. \tag{6}$$

As shown in Materials and methods, this equation yields a stable steady-state length $L_{ss}$ for the single flagellum.

## Limiting-pool mechanisms alone cannot account for the rapid length equalization observed in severing experiments

Whereas for a single flagellum the limiting-pool mechanism leads to a stable steady-state length, we show next that this mechanism is insufficient for the simultaneous length control of two flagella. A general limiting-pool model for the dynamics of two flagella having lengths $L_1(t)$ and $L_2(t)$ is given by the following ODEs:

$$\frac{\mathrm{d}L_1}{\mathrm{d}t} = \gamma J_1 T_{f,1} - d, \tag{7}$$

$$\frac{\mathrm{d}L_2}{\mathrm{d}t} = \gamma J_2 T_{f,2} - d, \tag{8}$$

where $J_i$ and $T_{f,i}$ for $i = 1, 2$ denote the fluxes of IFT particles into the two flagella and the amounts of free tubulin in their basal pools, respectively. As we will show next, the particular forms of $J_i$ and $T_{f,i}$ depend on how the pools are coupled. In particular, the fluxes and free tubulin will be equal for the two flagella if the basal proteins are held in a common shared pool.

Severing experiments illustrate that the flagella are coupled. We consider various modes of coupling, that is shared or separate motor pools and shared or separate tubulin pools, as depicted in *Figure 3a*, that give rise to different forms of $J_i$ and $T_{f,i}$ within our model. We investigate their consequences for length control by focusing on the solutions to the steady-state equations

$$0 = \gamma J_1 T_{f,1} - d, \tag{9}$$

$$0 = \gamma J_2 T_{f,2} - d. \tag{10}$$

Length control implies that these steady-state equations must yield a unique steady-state solution for $L_1$ and $L_2$. We consider the steady-state lengths before and after severing. It is known from experiments (*Figure 1b*) that initially the two flagella have equal lengths, and that after severing, which is accompanied by a loss of material (e.g. tubulin and motors lost from the severed flagellum), there is a rapid equalization of flagellar lengths. This initial, fast equalization of lengths leads to flagella that are shorter than they were before the severing. These experimental observations may be used to reject candidate models. Here we focus on the short-time dynamics after severing and therefore do not account for protein replenishment; however, as we show in the next section, incorporating protein replenishment is incompatible with the constant disassembly models considered here.

As we shall see, the constant disassembly models within our model space do not yield the rapid length equalization observed experimentally. As shown in *Figure 3b* and considered next on a case-

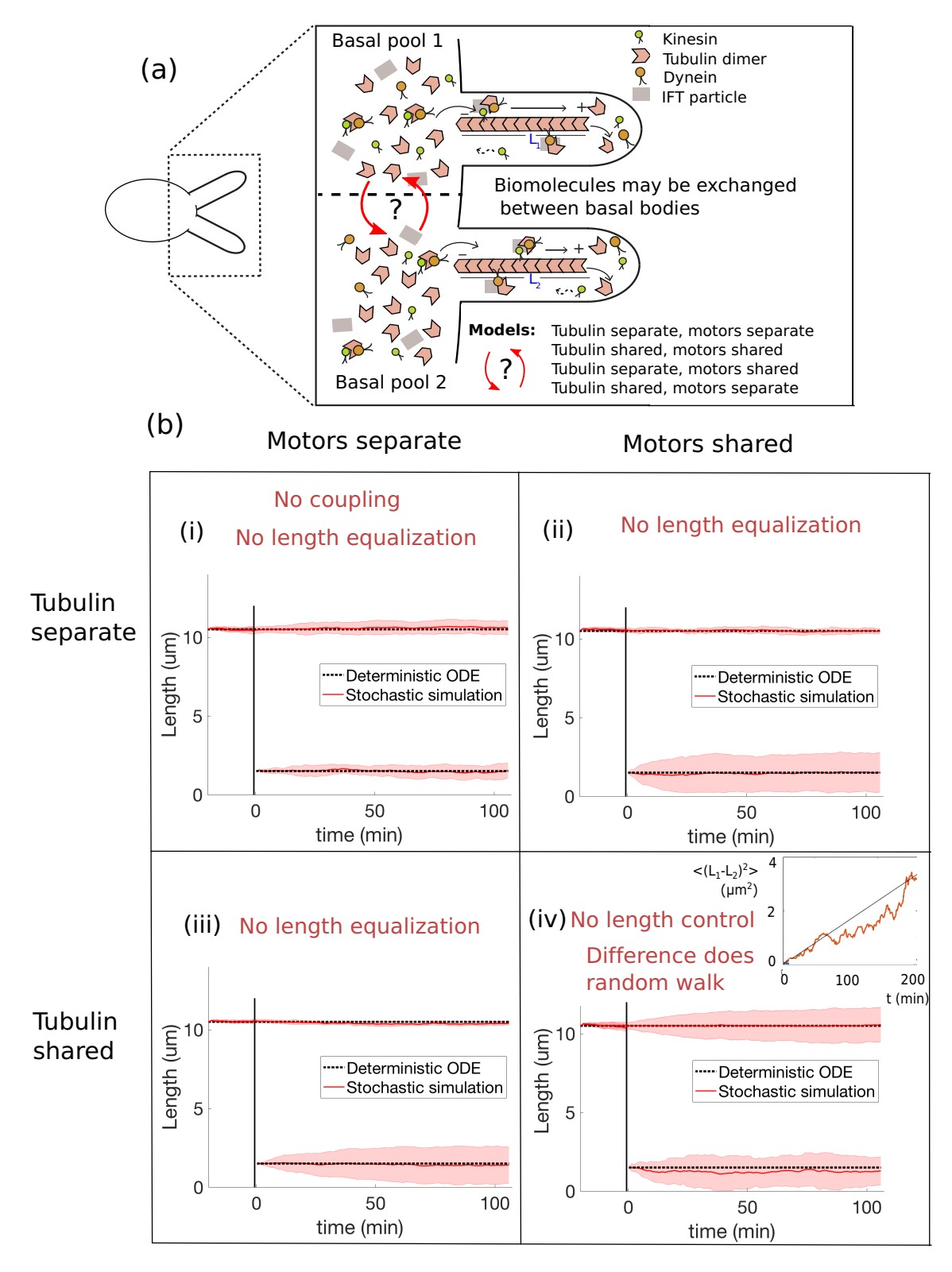

**Figure 3.** Length dynamics of two flagella assembling from shared pools of building blocks. (**a**) Flagellar assembly in *Chlamydomonas reinhardtii* and modes of coupling between basal proteins pools. (**b**) Simulations of the severing experiment using different modes of coupling between basal pools of proteins. In all cases severing occurs at time zero. (**i**) In the case of separate pools of both motors and tubulin (*Equations 44 and 45*), the unsevered flagellum does not decrease in length. (**ii**) When only motors are shared (*Equations 49 and 50*), the flagellar lengths do not equalize after severing. (**iii**)

*Figure 3 continued on next page*

*Figure 3 continued*

When only tubulin is shared (*Equations 56 and 57*), the flagellar lengths do not equalize after severing. (iv) When both tubulin and motors are shared (*Equations 58 and 59*), there is no unique steady-state. The difference between the two flagellar lengths undergoes a random walk, as shown by the mean square displacement (inset, average over 10 runs).

The online version of this article includes the following video(s) for figure 3:

**Figure 3—video 1.** The constant disassembly model with tubulin separate and motors shared does not yield length equalization.
https://elifesciences.org/articles/42599#fig3video1
**Figure 3—video 2.** The constant disassembly model with tubulin shared and motors separate does not yield length equalization.
https://elifesciences.org/articles/42599#fig3video2
**Figure 3—video 3.** The constant disassembly model with both tubulin and motors shared does not yield length control.
https://elifesciences.org/articles/42599#fig3video3
**Figure 3—video 4.** Replenishing protein pools in the constant disassembly model with tubulin separate and motors shared does not yield length control.
https://elifesciences.org/articles/42599#fig3video4
**Figure 3—video 5.** Replenishing protein pools in the constant disassembly model with tubulin shared and motors separate does not yield length control.
https://elifesciences.org/articles/42599#fig3video5
**Figure 3—video 6.** Replenishing protein pools in the constant disassembly model with both tubulin and motors shared does not yield length control.
https://elifesciences.org/articles/42599#fig3video6

by-case basis, if the lengths are in steady-state prior to severing, they are also in steady-state immediately after severing as well so that no length equalization occurs. This is because in these models the assembly and disassembly rates depend only on free protein levels in the basal protein pools, which are unaffected by severing. To be more precise, we may rewrite the governing equations *Equations 7 and 8* in terms of the number of free motors $M_{f,1}$ and $M_{f,2}$ in the following form:

$$\frac{\mathrm{d}L_1}{\mathrm{d}t} = \gamma k_{\mathrm{on}} M_{f,1} T_{f,1} - d, \tag{11}$$

$$\frac{\mathrm{d}L_2}{\mathrm{d}t} = \gamma k_{\mathrm{on}} M_{f,2} T_{f,2} - d. \tag{12}$$

In order to simulate a severing experiment, we first allow the two flagella to reach steady-state. Prior to severing, there is an amount of tubulin $L_1$ and a number of motors $M_{1,b} + M_{1,d}$ loaded on the first flagellum corresponding to the numbers of motors undergoing ballistic or diffusive motion (see Materials and methods). Severing is modeled by setting $L_1 \to 0$, $T_1 \to (T_1 - L_1)$, and $M_1 \to (M_1 - M_{1,b} - M_{1,d})$. Because severing does not change any of the protein levels in the basal pools, $M_{f,i}$ and $T_{f,i}$ for $i = 1, 2$ are the same before and after severing. Moreover, the linear concentration profile of diffusing motors remains linear in the truncated flagellum; no redistribution of motors is required to maintain quasi-steady state. As a consequence the constant disassembly models predict no length recovery of the severed flagellum, contrary to what is observed.

## Tubulin separate, motors separate
The case of separate tubulin pools and individual motor pools leads to two uncoupled instances of (6). This is fundamentally inconsistent with the coupling observed in severing experiments, in particular the significant decrease in the length of the unsevered flagellum (*Figure 3b(i)*). In this model, the unsevered flagellum does not change length after severing, and therefore it may be ruled out.

## Tubulin separate, motors shared
When motors are shared through a common pool, IFT particles are injected into either flagellum with equal probability. Therefore $J_1 = J_2 =: J$ and it can be shown by a straightforward generalization of *Equation 4* (see Materials and methods) that the flux satisfies

$$J = \frac{k_{\mathrm{on}}M/2}{1 + k_{\mathrm{on}}(L_1 + L_2)/2v + k_{\mathrm{on}}(L_1^2 + L_2^2)/4D}. \tag{13}$$

Initially the tubulin pools are equal, as are the flagellar lengths. However, after the loss of material due to severing, $T_1 \neq T_2$ and the lengths do not equalize (see *Figure 3b(ii)* and *Figure 3—video 1*).

Because of the separate tubulin pools, severing leads to asymmetrical tubulin depletion and unequal steady-state lengths after severing. Therefore, we may rule out the model.

## Tubulin shared, motors separate

This case is analogous to the previous model in which only motors are shared, but this time only the tubulin pools are shared so that $T_{f,1} = T_{f,2} =: T_f$, with

$$T_f = T - L_1 - L_2. \tag{14}$$

As shown in *Figure 3b(iii)* and *Figure 3—video 2*, this model does not capture the length equalization observed in severing experiments and therefore may be ruled out. See Materials and methods for details.

## Tubulin shared, motors shared

The flux resulting from the shared motor assumption is the same as in *Equation 13*, and we are left with the steady-state equations

$$0 = \gamma J(T - L_1 - L_2) - d, \tag{15}$$

$$0 = \gamma J(T - L_1 - L_2) - d. \tag{16}$$

Because the two steady-state equations are identical, that is $\mathrm{d}L_1/\mathrm{d}t = \mathrm{d}L_2/\mathrm{d}t$, this model does not account for the simultaneous positive and negative growth rates for the two flagella observed in the severing experiment.

More strikingly, subtracting the steady-state equations yields

$$\frac{\mathrm{d}(L_1 - L_2)}{\mathrm{d}t} = 0, \tag{17}$$

so that the difference in lengths is not controlled at all. Note that this result is *independent* of the parameters. Indeed, a similar conclusion was reached in the context of actin filaments (*Mohapatra et al., 2017*), in which it was observed that sharing all biomolecules between filaments does not yield simultaneous length control. In the context of the full stochastic simulations, this degeneracy is manifested by the difference in lengths undergoing a random walk (inset to *Figure 3b (iv)* and *Figure 3—video 3*).

The above analysis shows that, regardless of the manner in which tubulin and motors are shared between flagella, the constant disassembly models we have considered are unable to explain the results of severing experiments. Although sharing either tubulin or motors, but not both, yields a unique steady-state, these models do not agree with the rapid length equalization observed in severing experiments. This motivates us to extend our study beyond the models considered thus far.

## Controlling protein levels in the basal pool is incompatible with the constant disassembly models considered

So far, we have assumed that the tubulin pool $T$ and motor pool $M$ are fixed throughout the simulation. While this assumption is reasonable for the fast initial phase of the severing experiment in which the flagellar lengths rapidly equalize, the slower second phase of recovery to the original steady state lengths requires replenishment of proteins back to their original levels. This was shown experimentally by using cycloheximide at the time of severing to block the synthesis of new proteins (*Rosenbaum et al., 1969*), resulting in shorter flagella that did not recover to their original lengths.

It may seem plausible that adding an external control mechanism that replenishes protein levels would lead to length equalization, thus resolving the issue of unequal steady-state lengths after severing. However, as we next show, adding such a control mechanism on free proteins levels in the basal pool does not lead to length equalization. Instead, in this case the constant disassembly models we have considered completely fail to control lengths.

To incorporate control on protein levels into our model, we assume that the total levels $T$ and $M$ of tubulin and motors are replenished over a timescale $\tau_r$ as the cell synthesizes new protein to achieve target protein levels $\overline{T}_f$ and $\overline{M}_f$ in the basal pool:

$$\tau_r \frac{\mathrm{d}T}{\mathrm{d}t} = \overline{T}_f - T_f, \tag{18}$$

$$\tau_r \frac{\mathrm{d}M}{\mathrm{d}t} = \overline{M}_f - M_f. \tag{19}$$

In steady-state, the above equations become $T_f = \overline{T}_f$ and $M_f = \overline{M}_f$. The steady-state equation for length then becomes $0 = \gamma k_{\mathrm{on}} \overline{M}_f \overline{T}_f - d$, that is length drops out of the assembly term completely! Therefore, this external control mechanism actually *destabilizes* the flagellar lengths (see *Figure 3— videos 4—6* to observe this destabilization for various modes of coupling). Note that although we have used the simplest case of linear feedback *Equations 18 and 19* to illustrate the point, this argument is general and does not depend on the details of the control mechanism. The only requirement is that the free protein levels $T_f$ and $M_f$ are driven to their target values $\overline{T}_f$ and $\overline{M}_f$ in steady state. This argument gives another compelling reason to look beyond the constant disassembly models considered thus far.

## Tubulin shared, motors shared and concentration-dependent disassembly

We next consider a model that allows for full exchange of IFT components between basal protein pools and replaces the constant disassembly assumption with a concentration-dependent disassembly rate. The assumption of a constant disassembly rate was based on experiments on mutants in which IFT was disabled (*Marshall et al., 2005*). However, subsequent experiments in organisms with intact IFT led to 50-fold greater disassembly rates than those measured in the absence of IFT (*Ludington et al., 2012*).

Experimental observations that some kinesin species (e.g. kinesin-13) participate in microtubule disassembly (*Piao et al., 2009*) provide a potential biochemical basis for IFT-dependent disassembly. In what follows we take the disassembly rate to depend on the concentration of a depolymerizing protein. This is a reasonable model for a depolymerizer that is non-processive in its depolymerization activity in that it removes at most a few tubulin subunits before falling off into a deactivated state. (The case of processive depolymerizers is investigated in Appendix 3).

We will assume that the depolymerizer has the same motion as kinesin-2—uninterrupted ballistic motion to the tip followed by diffusive motion to the base—resulting in a linear concentration profile. This would be the case for any non-motile protein that is transported ballistically to the flagellar tip as IFT cargo and diffuses back to the flagellar base. Note however that the ballistic-to-diffusive assumption is not essential for the model; so long as there is a gradient in the depolymerizer concentration, the conclusion of simultaneous length control holds. The essential ingredient is the non-constant concentration along the length of the flagellum, which in this case is achieved by ballistic anterograde motion and diffusive return.

Given that the formulas for the steady-state flux apply for any rate-limiting IFT protein undergoing ballistic-to-diffusive motion along the flagella, for convenience we assume in what follows that the depolymerizer is the rate-limiting protein. However, this assumption is made only for convenience; in the more general case that the depolymerizer and the rate-limiting IFT protein are different, the same results are obtained with suitably rescaled parameters, as shown in Appendix 3.

We replace the assumption of constant disassembly by a disassembly speed of the form $d_0 + d_1 c_d(L)$, where $c_d(L)$ is the concentration of diffusing motors at the tip of the flagellum and $d_1 > 0$. In general the disassembly rate may be an arbitrary function of concentration, in which case this model may be viewed as a first-order Taylor series expansion valid near steady-state. The flux and concentration at the flagellar tip are related by $c_d(L) = JL/D$ (*Equation 31* in Materials and methods and *Figure 2b (inset)*), so that we may rewrite the governing equations as

$$\frac{\mathrm{d}L_1}{\mathrm{d}t} = \gamma J(T - L_1 - L_2) - d_0 - d_1 \frac{JL_1}{D}, \tag{20}$$

$$\frac{\mathrm{d}L_2}{\mathrm{d}t} = \gamma J(T - L_1 - L_2) - d_0 - d_1\frac{JL_2}{D},$$

(21)

where as before in the case of shared motors

$$J = \frac{k_{\mathrm{on}}M/2}{1 + k_{\mathrm{on}}(L_1 + L_2)/2v + k_{\mathrm{on}}(L_1^2 + L_2^2)/4D}.$$

(22)

This model yields simultaneous length control and length equalization after severing (*Figure 4* and *Figure 4—video 1*). Subtracting *Equation 21* from *Equation 20*, it follows immediately that $L_{1,ss} = L_{2,ss} =: L_{ss}$, and solving for the steady-state length results in

$$L_{ss} = \left(\frac{D}{v} + \frac{\gamma DM}{d_0} + \frac{Md_1}{2d_0}\right)\left(-1 + \sqrt{1 + \left(\frac{d_0T}{\gamma MD}\right)\frac{1 - 2d_0/\gamma k_{\mathrm{on}}MT}{(1 + d_0/\gamma Mv + d_1/2\gamma D)^2}}\right).$$

(23)

In Appendix 3 we show that this solution is stable using linear stability analysis. Therefore, concentration-dependent disassembly yields simultaneous length control when all biomolecules are shared between flagella and is consistent with the rapid length equalization observed after severing *Figure 4b*. The presence of a concentration gradient is a critical ingredient in this model and here it is achieved by ballistic transport to the flagellar tip with diffusive return. The concentration gradient makes the disassembly rates length-dependent and yields independent equations for the steady-state lengths.

Unlike the constant disassembly models we have considered, for which a limiting-pool mechanism is essential for length control, concentration-dependent disassembly yields length control under mild assumptions including the case that all biomolecules are in excess. Nevertheless, limiting-pools of biomolecules are needed to capture the depletion effects observed in severing experiments on *Chlamydomonas*, for example the shortening of the unsevered flagellum and the previously-mentioned absence of length recovery after cyclohexamide treatment (*Rosenbaum et al., 1969*). In Appendix 3 we explain that limiting pools of IFT motors are necessary for agreement with data whereas tubulin may either be limited or in excess. The presence of a limiting-pool once again raises the question, now in the context of concentration-dependent disassembly, of how biomolecules may be shared between flagella. In Appendix 3 we show that all relevant biomolecule pools must be shared for the concentration-dependent disassembly model to capture the rapid length equalization observed.

Another feature of the concentration-dependent disassembly model is that it allows for an external control mechanism on protein levels in the basal pool, unlike the constant disassembly models we have considered. As shown in *Figure 4c* and *Figure 4—video 2*, upon including protein replenishment via *Equations 18 and 19* on a timescale of $\tau_r = 10$ mins, the recovery of the flagella back to their original lengths is in reasonable agreement with experimental data.

## Generalization to $N > 2$ flagella

The concentration-dependent disassembly model may be generalized to arbitrary flagellar number $N$, and here we demonstrate simultaneous length control in the case of $N = 8$ flagella (*Figure 5*).

Because motors are shared, the injection fluxes are equal and $J_i = J$ for all $i = 1\ldots N$, with $J$ satisfying

$$J = \frac{k_{\mathrm{on}}M/N}{1 + k_{\mathrm{on}}(\sum_{i=1}^N L_i)/Nv + k_{\mathrm{on}}(\sum_{i=1}^N L_i^2)/2ND},$$

(24)

and length dynamics given by

$$\frac{\mathrm{d}L_i}{\mathrm{d}t} = \gamma J\left(T - \sum_{j=1}^N L_j\right) - d_0 - d_1\frac{JL_i}{D}, \quad i = 1\ldots N,$$

(25)

where we have applied the boundary condition $c_{d,i}(0) = 0$ as before. Taking any pairwise difference between the $i^{\mathrm{th}}$ and $j^{\mathrm{th}}$ equations at steady-state yields immediately $L_{i,ss} = L_{j,ss}$, so that the steady-state lengths are equal to

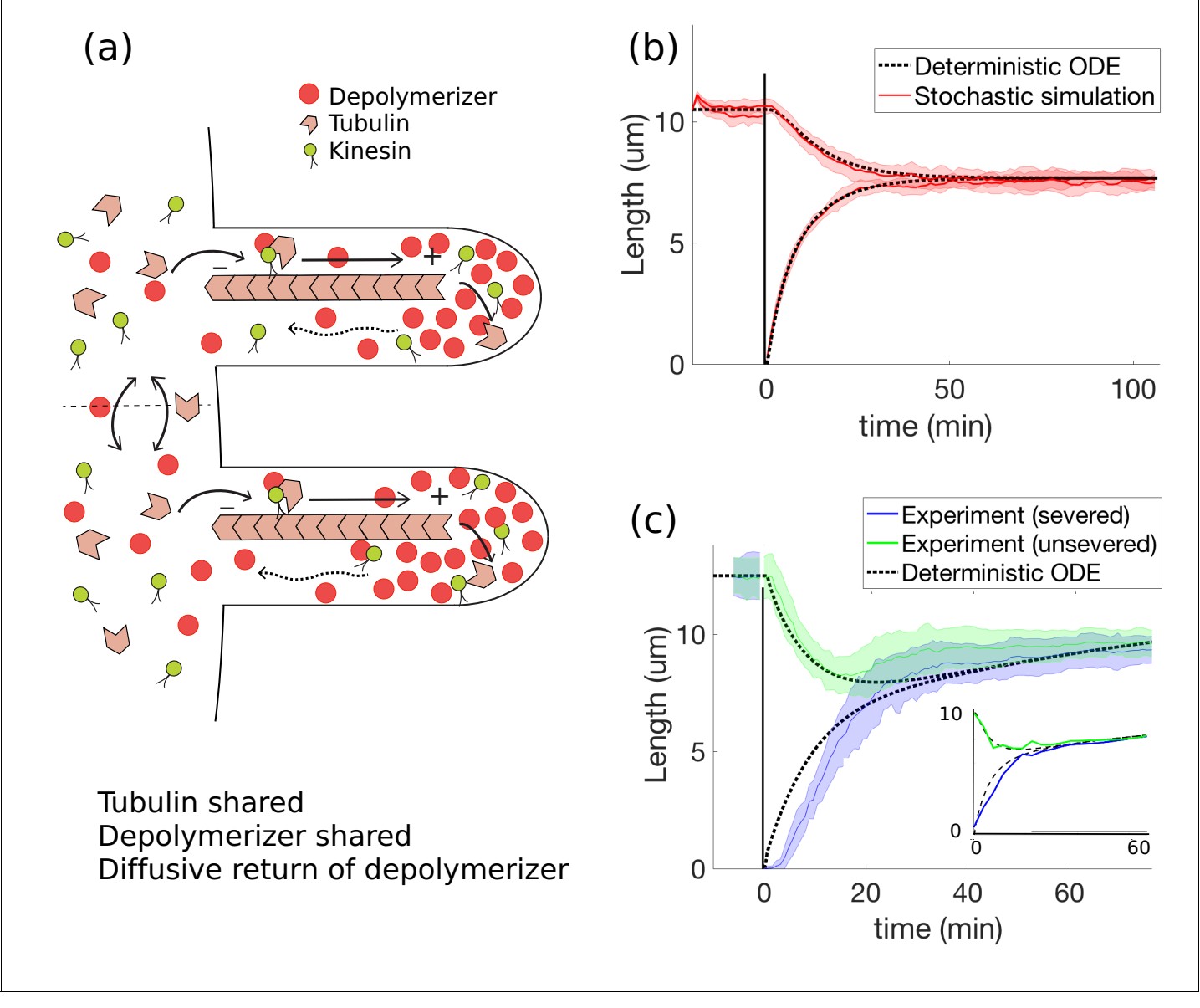

**Figure 4.** Concentration-dependent disassembly model: simultaneous length control is achieved using shared tubulin and shared depolymerizers. (a) The depolymerizer moves ballistically to the flagellar tip and diffuses back, (b) The model *Equations 20 and 21* captures rapid length equalization, (c) Protein replenishment with timescale $\tau_r = 5\,\text{min}$ is included through *Equations 18 and 19* and the model is fit to severing data from *Ludington et al. (2012)* (and to *Rosenbaum et al., 1969*, see inset). For the fit to *Ludington et al. (2012)*, we use fitting parameters of $L_{ss} = 12.5\,\mu\text{m}$, $T = 28\,\mu\text{m}$, $d_0 = 2\,\mu\text{m/min}$, $d_1 = 2.7 \times 10^{-3}\,\mu\text{m}^2/\text{min}$, and $\gamma k_{\text{on}} M = 2.5\,\text{min}^{-1}$; all other parameters are as in *Table 1*. For the fit to *Rosenbaum et al. (1969)*, we use the same fitting parameters except for $\tau_r = 8\,\text{min}$, $L_{ss} = 10.2\,\mu\text{m}$, and $d_1 = 8.4 \times 10^{-3}\,\mu\text{m}^2/\text{min}$.

The online version of this article includes the following video(s) for figure 4:

**Figure 4—video 1.** The active disassembly model with both tubulin and motors shared but without protein replenishment exhibits rapid length equalization.

https://elifesciences.org/articles/42599#fig4video1

**Figure 4—video 2.** Replenishing protein pools in the active disassembly model with both tubulin and motors shared exhibits rapid length equalization and slow recovery.

https://elifesciences.org/articles/42599#fig4video2

$$L_{i,ss} = \left(\frac{D}{v} + \frac{\gamma DM}{d_0} + \frac{Md_1}{Nd_0}\right)\left(-1 + \sqrt{1 + \left(\frac{2d_0 T}{N\gamma MD}\right)\frac{1 - Nd_0/\gamma k_{\mathrm{on}}MT}{(1 + d_0/\gamma Mv + d_1/N\gamma D)^2}}\right), \tag{26}$$

for all $i = 1,\ldots,N$. Stability follows from analyzing the linearized equations, as shown in Appendix 3.

## Discussion

In this work, we capture the aspects of IFT essential for length control, that is the motor-driven transport of tubulin across the flagellum, to explore models of flagellar length dynamics. Our theoretical framework makes it possible to investigate the consequences of biomolecule exchange between flagella on length control. In our initial exploration, in which we take the disassembly rate to be constant, we find that sharing both tubulin and motors leads to an indeterminate system of equations regardless of the details of the model, whereas sharing either tubulin or motors, but not both, results in simultaneous length control of both flagella. However, by examining the steady-state lengths immediately before and after severing and accounting for depletion in both the tubulin and motor pools, we observe that none of the constant disassembly models we have considered are able to capture the length equalization observed in experiments.

Given that the constant disassembly models we have considered are unable to explain the experiments, we have proposed a model in which disassembly depends on the local concentration of a depolymerizing protein at the tip of the flagellum (*Figure 4*). In this 'active disassembly' model, a length-dependent concentration at the tip is achieved by assuming that the depolymerizer undergoes ballistic motion to the tip and returns diffusively. This model agrees with the results of severing experiments. As suggested by the title of *Hendel et al. (2018)*, the linear concentration gradient generated by diffusion acts as a ruler. However, in our model diffusion must be combined with a mechanism such as concentration-dependent disassembly; the constant disassembly models we have considered are inconsistent with severing experiments regardless of whether the motor dynamics are ballistic, diffusive, or some combination.

We remark on the differences between our model and *Hendel et al. (2018)*, in which simultaneous length control was obtained using a balance point model with constant disassembly. As we describe in Materials and methods, the formulation of *Hendel et al. (2018)* is very similar to *Equations 56 and 57* derived in the case of shared tubulin pools and separate motor pools with constant disassembly. Whereas these equations do not lead to length equalization after severing in the case of no protein replenishment (*Figure 3b(iii)*), the model of *Hendel et al. (2018)* achieves length equalization by replenishing the total number of motors on the flagellum, that is through an additional control mechanism that instantaneously adds the motors lost through severing back into the basal pool. The importance of replenishment for the model appears to be inconsistent with severing experiments that show length equalization occurs even when protein synthesis is blocked using cyclohexamide (*Rosenbaum et al., 1969*). In contrast, the concentration-dependent disassembly model achieves length equalization with or without replenishment (*Figure 4b* and *Figure 4c*). Note further that controlling motor number in the flagellum is not equivalent to controlling protein concentrations in the basal pool. As shown in Results, when basal pool concentrations are controlled according to *Equations 18 and 19*, there is a breakdown of length control for the constant disassembly models.

Our model predicts that the tubulin and depolymerizer pools must both be shared for the concentration-dependent disassembly model to capture the rapid length equalization observed

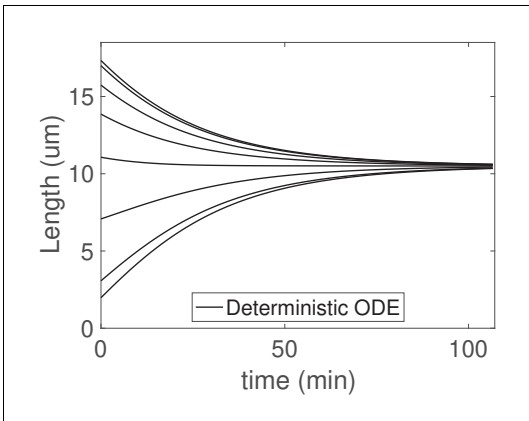

**Figure 5.** The concentration-dependent disassembly model generalizes to arbitrary flagellar number $N$. We solve (*Equations 25*) with $N = 8$ flagella and the larger shared pool $T = 336\,\mu\mathrm{m}$, $d_0 = 0.1\,\mu\mathrm{m/min}$, and $d_1 = 120\,\mu\mathrm{m}^2/\mathrm{min}$; otherwise all parameters are as in *Table 1*.

(Appendix 3). This illustrates the dramatic consequences in behavior that can occur when biomolecules are shared between compartments, and highlights the importance of knowing which proteins are exchanged between the basal pools in the context of flagellar length control. In particular, having a protein that is *not* exchanged can provide simultaneous length control by a limiting-pool mechanism, but it introduces asymmetries that contradict the length equalization observed in severing experiments.

In addition to our claim that the depolymerization rate is non-constant and dependent on length, our model leads to testable predictions that may be useful in identifying candidate depolymerizers. For example, according to our model the depolymerizer active in length control is not uniformly distributed along the flagellum; its concentration increases toward the flagellar tip. This could be tested experimentally by fluorescently labeling candidate depolymerizers and studying their concentration profiles along the flagellum as recently done to characterize the concentration profile of kinesin-13 in *Giardia* (*McInally et al., 2019*).

In our model the essential ingredient that leads to simultaneous length control is the presence of a depolymerizing protein with a concentration gradient along the flagellum. How proteins can develop and maintain such concentration profiles is therefore one of the key questions raised by the model. Indeed, concentration gradients (unassociated with depolymerizing activity) were observed already in *Hendel et al. (2018)* as the result of ballistic-to-diffusive motion. Although here as proof of principle this concentration gradient is achieved by the mechanism of ballistic-to-diffusive motion, our main results are independent of the detailed form of this concentration gradient and how it is generated. As shown in *Appendix 3—figure 3b*, our model allows for depolymerizers with nonlinear concentration profiles and different patterns of motion, for example exponential concentration distributions such as those recently observed in *Giardia* (*McInally et al., 2019*) and those generated by motile proteins that bind and unbind to cytoskeletal filaments, as theorized in the context of actin-myosin systems (*Naoz et al., 2008*; *Orly et al., 2014*; *Pinkoviezky and Gov, 2014*; *Pinkoviezky and Gov, 2017*; *Yochelis et al., 2015*).

Our main results do not rely on many of the details of the model. Although here we have taken depolymerase activity to depend linearly on concentration, the model generalizes to the non-linear case in a straightforward manner. For a local depolymerization rate that is an arbitrary function of concentration, a Taylor series expansion may be performed as in *Klein et al. (2005)* to obtain the corresponding linearized system discussed in Appendix 3. Further, whereas here we have explored the case of non-motile depolymerizers transported to the flagellar tip by IFT, motile proteins could in principle aggregate at the flagellar tip independent of IFT. Although we are not aware of any such examples in *Chlamydomonas*, interestingly in budding yeast the motile protein Kip3p has been shown to depolymerize microtubules in a length-dependent manner (*Varga et al., 2009*). The aggregation of motile depolymerizing proteins has also been demonstrated in previous theoretical studies of microtubule length control (*Klein et al., 2005*; *Johann et al., 2012*; *Reese et al., 2014*); note however that these previous works differed from our model of flagellar IFT in that they considered isolated microtubules surrounded by a constant concentration bath and/or significant steric interactions between motile proteins.

Finally, although in principle concentration-dependent disassembly yields length control even the case that all biomolecules are in excess, in *Chlamydomonas* severing experiments have shown that depletion effects are important (e.g. cyclohexamide treatment yields short flagella that do not return to their original steady-state lengths [*Rosenbaum et al., 1969*]). Within our model such depletion effects arise through limiting-pools of proteins, and in Appendix 3 we explain that limiting pools of IFT motors are necessary for agreement with data whereas tubulin may either be limited or in excess.

Although our results suggest that having a depolymerizer—one which is ballistically transported to the tip and then diffuses back—provides an appealing model for simultaneous length control, such a depolymeriser has yet to be identified in *Chlamydomonas*. Further experiments such as the single molecule turnaround experiments pioneered recently in *C. elegans* (*Mijalkovic et al., 2018*) are needed to establish the identity of the hypothesized depolymerizer. While recent experiments have shown kinesin-13 to be involved in length control in *Giardia* (*McInally et al., 2019*), the observation that only negligible amounts of flagellar kinesin-13 are present at steady-state (*Wang et al., 2013*) appears to preclude it from being the candidate depolymerizer of our model. Other candidates include aurora-like kinase CALK, which has been shown to influence disassembly through its

state of phosphorylation (*Luo et al., 2011*; *Cao et al., 2013*), and CNK2, a NIMA-related protein kinase known to localize to flagella (*Bradley and Quarmby, 2005*) whose absence yields Chlamydomonas with abnormally long flagella and decreased disassembly rates (*Hilton et al., 2013*).

On the side of theory, a promising avenue to further test the active disassembly model and discriminate between different possibilities is to study length fluctuations about steady-state. This could be done using agent-based stochastic simulations, stochastic differential equations, or a combination of the two. The fluctuation spectra of each model provides a signature that can be used to assess both the general model framework and to test the autocorrelation timescales predicted by each model (*Amir and Balaban, 2018*).

# Materials and methods

## Single flagellum

We first consider a single flagellum with time-dependent length $L(t)$. Given the highly regular structure of the axoneme revealed by cryo-electron microscopy (*Bui et al., 2008*; *Barber et al., 2012*), we assume a constant cross-sectional area in which case the flagellar geometry is fully described by its length. In our model the flagellar assembly rate is proportional to the flux $J$ of IFT particles times the tubulin carried per particle. The tubulin carried is proportional to the amount of free tubulin $T_f$, assuming mass action kinetics in the basal pool (i.e. constant probability per time of tubulin binding to an IFT particle). For now we take the disassembly speed to be equal to a constant $d$. This yields the growth rate

$$\frac{\mathrm{d}L}{\mathrm{d}t} = \gamma J T_f - d. \tag{27}$$

As described in Results, we assume that motors are conserved having total number $M = M_f + M_b + M_d$, where $M_f$ is the number of motors freely available in the basal protein pool, $M_b$ is the number of motors moving ballistically on the flagellum in IFT particles, and $M_d$ is the number of motors moving diffusively.

We assume a limiting pool of tubulin in addition to the limiting pool of motors, that is tubulin is conserved with total amount $T = T_f + L$. As the flagellum grows it incorporates more tubulin and the size of the free tubulin pool decreases. (In reality $T = T_b + T_f + L$, where $T_b$ is the amount of tubulin undergoing IFT, but this correction is negligible; the amount of tubulin moving ballistically in IFT satisfies $T_b/T_f < 2\gamma k_{\mathrm{on}} M L_{ss}/v$, and consequently $T_b/T_f < 2.6 \times 10^{-3}$ for the parameters contained in *Table 1*).

### Flux balance

In our model the flux, or injection rate, is proportional to the number of free molecular motors $M_f$ so that

$$J = k_{\mathrm{on}} M_f, \tag{28}$$

according to mass action kinetics with first-order rate constant $k_{\mathrm{on}}$. By mass action and conservation of motors, the flux of motors may be expressed as

$$J = k_{\mathrm{on}}(M - M_b - M_d). \tag{29}$$

The ballistic flux is related to concentration in a simple manner. It satisfies $J = \overline{c_a} v$, where $\overline{c_a}$ is the average concentration of motors moving in the anterograde direction and $v$ is the anterograde velocity. (As mentioned previously, it follows from the quasi-steady state assumption that the injection flux, anterograde flux, and retrograde flux are equal so that there is a single flux $J$.) Therefore

$$M_b = L\overline{c_a} = LJ/v. \tag{30}$$

In quasi-steady state, the diffusive flux $D\partial c_d/\partial x$ must equal the injection rate $J$. By Fick's law, for constant $J$ the concentration profile $c_d(x)$ of the diffusing motors is linear, that is

$$c_d(x) = \overline{c_d} + \frac{J}{D}\left(x - \frac{L}{2}\right), \tag{31}$$

in which $\overline{c_d}$ is the average concentration along the flagellum and $D$ is the diffusion constant (*Figure 2b (inset)*). We treat the flagellar base as a diffusive sink by fixing the boundary condition $c_d(0) = 0$, which assumes that motors in the basal pool cannot leak diffusively into the flagellum; instead they attach to the microtubules in the axoneme and move directionally toward the tip (More general boundary conditions are discussed in Appendix 2.) This implies that

$$\overline{c_d} = \frac{JL}{2D}. \tag{32}$$

Therefore

$$M_d = \overline{c_d}L = \left(\frac{JL}{2D}\right)L = \frac{JL^2}{2D}, \tag{33}$$

for the diffusively-moving motors. Substituting the expressions *Equation 30* and *Equation 33* for $M_b$ and $M_d$ into *Equation 29* results in

$$J = \frac{k_{\mathrm{on}}M}{1 + k_{\mathrm{on}}L/v + k_{\mathrm{on}}L^2/2D}. \tag{34}$$

The denominator is a quadratic function in length, and it is interesting to note that the flux has a similar functional form to the familiar substrate production rate in Michaelis-Menten enzyme kinetics (*Fall, 2002*); this is because of the separation of timescales assumption invoked in both derivations.

Using the above expression for the flux in the growth rate *Equation 27* together with the relation $T_f = T - L$ gives

$$\frac{\mathrm{d}L}{\mathrm{d}t} = \frac{\gamma k_{\mathrm{on}}M}{1 + k_{\mathrm{on}}L/v + k_{\mathrm{on}}L^2/2D}(T - L) - d. \tag{35}$$

Solving *Equation 35* for the steady-state results in a quadratic equation for $L_{ss}$. One root is always negative, leaving the solution

$$L_{ss} = \left(\frac{D}{v} + \frac{\gamma DM}{d}\right)\left(-1 + \sqrt{1 + \left(\frac{2dT}{\gamma MD}\right)\frac{1 - d/\gamma k_{\mathrm{on}}MT}{(1 + d/\gamma Mv)^2}}\right), \tag{36}$$

which is positive provided that $T > d/\gamma kM$. (We remind the reader that terms such as $d/\gamma k_{\mathrm{on}}M$ are to be interpreted as $d/(\gamma k_{\mathrm{on}}M)$.) This inequality provides a theoretical lower limit on the product of total motors and tubulin needed to obtain a positive steady-state length. When the inequality is not satisfied, that is $T \leq d/\gamma k_{\mathrm{on}}M$, the disassembly term dominates and the length shrinks to zero.

We evaluate the stability of this solution by linearizing about $L_{ss}$. Expanding to first order in $\Delta L := L - L_{ss}$, we find that the steady-state is stable, that is $\mathrm{d}(\Delta L)/\mathrm{d}t = -\lambda(\Delta L)$ with $\lambda$ a positive constant given by

$$\lambda = \left(\frac{\gamma k_{\mathrm{on}}M}{1 + k_{\mathrm{on}}L_{ss}/v + k_{\mathrm{on}}L_{ss}^2/2D}\right)\left(1 + \frac{k_{\mathrm{on}}/v + k_{\mathrm{on}}L_{ss}/D}{1 + k_{\mathrm{on}}L_{ss}/v + k_{\mathrm{on}}L_{ss}^2/2D}(T - L_{ss})\right). \tag{37}$$

Based on the parameters estimated in Appendix 2, the associated timescale $\tau := 1/\lambda$ is approximately 15 min, which is consistent with experiment. This timescale is long compared to the few tens of seconds needed for molecular motors to traverse the flagellum in IFT, which justifies a posteriori our approximation of IFT as a quasi-steady state process.

We next consider the parameter space associated with the length dynamics. Introducing the non-dimensional length $\widetilde{L} = L/L_{ss}$ and nondimensional time $\widetilde{t} = t\gamma k_{\mathrm{on}}MT/L_{ss}$, we may rewrite *Equation 35* in terms of the dimensionless parameters $\pi_1 = d/\gamma k_{\mathrm{on}}MT$, $\pi_2 = L_{ss}/T$, $\pi_3 = k_{\mathrm{on}}L_{ss}/v$, and $\pi_4 = k_{\mathrm{on}}L_{ss}^2/2D$ as

$$\frac{d\widetilde{L}}{d\widetilde{t}} = \frac{1 - \pi_2\widetilde{L}}{1 + \pi_3\widetilde{L} + \pi_4\widetilde{L}^2} - \pi_1. \tag{38}$$

We interpret these parameters as follows: $\pi_1$ is the ratio of disassembly and assembly rates, $\pi_2$ is the fraction of the tubulin pool taken up by the flagellum at steady-state, $\pi_3 = \tau_b/\tau_i$ is the ratio of the ballistic timescale $\tau_b := L_{ss}/v$ of IFT transport to the injection timescale $\tau_i := k_{on}^{-1}$, and $\pi_4 = \tau_d/\tau_i$ is an analogous ratio of the diffusive timescale $\tau_d = L_{ss}^2/2D$ to the injection timescale. (We could equivalently think of $\pi_3$ and $\pi_4$ as ratios of lengthscales related to the same physical processes.)

In terms of the experimentally measured parameters and those estimated in Appendix 2, we find $\pi_1 \approx 0.4$, $\pi_2 \approx 0.2$, $\pi_3 \approx 0.1$, and $\pi_4 \approx 0.8$. The relatively small values of $\pi_2$ and $\pi_3$ lead us to consider the limit $\pi_2 \to 0$ (i.e. no tubulin depletion) and $\pi_3 \to 0$ (i.e. instantaneous ballistic motion). In this limit, we have

$$\frac{dL}{dt} = \frac{\gamma k_{on}MT}{1 + k_{on}L^2/2D} - d, \tag{39}$$

nearly recovering the model of *Hendel et al. (2018)* with the distinction that, as mentioned above, in our model there is an additional constant term in the denominator. Note however that the essential difference between our model and *Hendel et al. (2018)* lies in their effective control mechanism on the number of motors loaded on the flagella, which is not captured by any differences in these formulas (see Discussion).

## Two flagella
In the case of two flagella with lengths $L_1(t)$ and $L_2(t)$ the length dynamics are given by

$$\frac{dL_1}{dt} = \gamma J_1 T_{f,1} - d, \tag{40}$$

$$\frac{dL_2}{dt} = \gamma J_2 T_{f,2} - d, \tag{41}$$

where $J_i$ and $T_{f,i}$ for $i = 1, 2$ denote the fluxes and free amounts of tubulin for the two flagella, which may be equal when the biomolecule pools are shared. We consider various modes of coupling between the flagella giving rise to different forms of $J_i$ and $T_{f,i}$ and their consequences for length control. To assess whether a model achieves simultaneous length control we analyze the stability of solutions to the following steady-state equations:

$$0 = \gamma J_1 T_{f,1} - d, \tag{42}$$

$$0 = \gamma J_2 T_{f,2} - d. \tag{43}$$

Here, we focus on short-time behavior, that is whether a candidate model yields rapid length equalization, and do not include the protein replenishment that takes place over a longer timescale.

### Tubulin separate, motors separate
The presence of separate tubulin pools and separate motor pools leads to two uncoupled instances of the single flagellum dynamics, that is

$$\frac{dL_1}{dt} = \frac{\gamma k_{on}M_1}{1 + k_{on}L_1/v + k_{on}L_1^2/2D}(T_1 - L_1) - d, \tag{44}$$

$$\frac{dL_2}{dt} = \frac{\gamma k_{on}M_2}{1 + k_{on}L_2/v + k_{on}L_2^2/2D}(T_2 - L_2) - d. \tag{45}$$

Setting $M_1 = M_2 = M$ and $T_1 = T_2 = T$ leads to steady state lengths given by *Equation 36*.

## Tubulin separate, motors shared

In the case of separate tubulin pools, we have $T_{f,1} = T_1 - L_1$ and $T_{f,2} = T_2 - L_2$. The flux may be calculated according to

$$J = \frac{1}{2}k_{on}M_f = \frac{1}{2}k_{on}(M - M_b - M_d),$$

(46)

where the factor of one-half comes from assuming equal injection probability into either flagellum. Further, $M_b = J(L_1 + L_2)/v$ and

$$M_d = \overline{c_{d,1}}L_1 + \overline{c_{d,2}}L_2 = J\left(\frac{L_1^2}{2D} + \frac{L_2^2}{2D}\right),$$

(47)

so that

$$J = \frac{k_{on}M/2}{1 + k_{on}(L_1 + L_2)/2v + k_{on}(L_1^2 + L_2^2)/4D}.$$

(48)

This yields the flagellar length dynamics

$$\frac{dL_1}{dt} = \frac{\gamma k_{on}M/2}{1 + k_{on}(L_1 + L_2)/2v + k_{on}(L_1^2 + L_2^2)/4D}(T_1 - L_1) - d,$$

(49)

$$\frac{dL_2}{dt} = \frac{\gamma k_{on}M/2}{1 + k_{on}(L_1 + L_2)/2v + k_{on}(L_1^2 + L_2^2)/4D}(T_2 - L_2) - d.$$

(50)

The steady-state equations are given by

$$0 = \gamma J(T_1 - L_{1,ss}) - d,$$

(51)

$$0 = \gamma J(T_2 - L_{2,ss}) - d.$$

(52)

When $T_1 = T_2$ it follows from subtracting the above equations that $L_{1,ss} = L_{2,ss}$. The steady-state equations are identical to the corresponding steady-state *Equation 35* for a single flagellum with $M$ replaced by $M/2$. Therefore the steady-state length satisfies *Equation 36* upon rescaling $M \to M/2$:

$$L_{1,ss} = L_{2,ss} = \left(\frac{D}{v} + \frac{\gamma DM}{2d}\right)\left(-1 + \sqrt{1 + \left(\frac{4dT}{\gamma MD}\right)\frac{1 - 2d/\gamma k_{on}MT}{(1 + 2d/\gamma Mv)^2}}\right).$$

(53)

The steady state lengths are only equal if $T_1 = T_2$, which is not the case after asymmetrical depletion of tubulin pools by severing (see *Figure 3b(ii)*).

## Tubulin shared, motors separate

We next consider the case in which tubulin is shared but the motor pools are separate. The separate motor pools yield decoupled fluxes identical to *Equations 44 and 45*:

$$J_1 = \frac{k_{on}M_1}{1 + k_{on}L_1/v + k_{on}L_1^2/2D},$$

(54)

$$J_2 = \frac{k_{on}M_2}{1 + k_{on}L_2/v + k_{on}L_2^2/2D},$$

(55)

which leads to the systems of equations

$$\frac{dL_1}{dt} = \frac{\gamma k_{on}M_1}{1 + k_{on}L_1/v + k_{on}L_1^2/2D}(T - L_1 - L_2) - d,$$

(56)

$$\frac{dL_2}{dt} = \frac{\gamma k_{on}M_2}{1 + k_{on}L_2/v + k_{on}L_2^2/2D}(T - L_1 - L_2) - d.$$

(57)

This system of equations is similar to the case of no sharing given by *Equations 44-45*, with the notable exception that the equations are coupled through the shared tubulin pool term $T - L_1 - L_2$. The resulting steady-state equations are identical to those of *Equation 35* for a single flagellum, with $T$ replaced by $T/2$ and $\gamma$ replaced by $2\gamma$. Therefore the steady-state lengths satisfy *Equation 36* upon rescaling $T \to T/2$ and $\gamma \to 2\gamma$. This model yields simultaneous length control, and the resulting steady-state lengths satisfy $L_{1,ss} = L_{2,ss}$ only if $M_1 = M_2 = M$. It follows that the steady-state lengths are unequal after severing because one of the two motor pools is depleted.

The model equations have a similar form to existing models (*Marshall et al., 2005*; *Hendel et al., 2018*), in which the assembly rates involve a factor of $T - L_1 - L_2$ and either a $1/L_i$ or $1/L_i^2$-dependence in the denominator, for $i = 1, 2$ as discussed earlier in the context of a single growing flagellum. Although the equations are similar, the absence of length equalization in our model (*Figure 3biii*) contrasts with the length equalization achieved in *Hendel et al. (2018)* by an additional control mechanism that instantaneously replenishes the number of motors on the flagellum after severing. As noted in the Discussion, the importance of protein replenishment for the model appears to be inconsistent with experimental results (*Rosenbaum et al., 1969*), which show that length equalization occurs even in the absence of new protein synthesis.

## Tubulin shared, motors shared

We finally consider the case in which both tubulin and motors are shared through a common pool. By the shared tubulin pool assumption $T_{f,1} = T_{f,2} = T - L_1 - L_2$. Further, by the shared motor pool assumption the injection rates satisfy $J_1 = J_2 \equiv J$. Therefore

$$\frac{dL_1}{dt} = \gamma J (T - L_1 - L_2) - d, \tag{58}$$

$$\frac{dL_2}{dt} = \gamma J (T - L_1 - L_2) - d, \tag{59}$$

in which $J$ satisfies *Equation 48*. We are left with the steady-state equations

$$0 = \gamma J (T - L_{1,ss} - L_{2,ss}) - d, \tag{60}$$

$$0 = \gamma J (T - L_{1,ss} - L_{2,ss}) - d. \tag{61}$$

These equations are identical, so that there is only a single equation for the two unknowns $L_{1,ss}$ and $L_{2,ss}$ and the steady-state lengths are indeterminate provided that the disassembly rate is constant. Note that this conclusion holds regardless of the particular form of the flux.

We next use linear stability analysis to demonstrate this breakdown of simultaneous length in greater detail. Let $L_{1,ss}$ and $L_{2,ss}$ denote any one of the infinitely-many possible solutions to *Equations 60 and 61*. Letting $\Delta L_1$ and $\Delta L_2$ be the deviations from steady-state such that $L_1 = L_{1,ss} + \Delta L_1$ and $L_2 = L_{2,ss} + \Delta L_2$, linearizing about any one of these solutions yields a matrix equation of the form

$$\frac{d}{dt} \begin{pmatrix} \Delta L_1 \\ \Delta L_2 \end{pmatrix} = - \begin{pmatrix} a & a \\ a & a \end{pmatrix} \begin{pmatrix} \Delta L_1 \\ \Delta L_2 \end{pmatrix}, \tag{62}$$

for $a > 0$. The $2 \times 2$ matrix above has an vanishing eigenvalue, as we now show. Diagonalizing in terms of the sum $\Sigma = \Delta L_1 + \Delta L_2$ and difference $\Gamma = \Delta L_1 - \Delta L_2$ gives

$$\frac{d}{dt} \begin{pmatrix} \Sigma \\ \Gamma \end{pmatrix} = - \begin{pmatrix} 2a & 0 \\ 0 & 0 \end{pmatrix} \begin{pmatrix} \Sigma \\ \Gamma \end{pmatrix}. \tag{63}$$

There is a vanishing eigenvalue associated to the difference of lengths, that is perturbations from steady-state in the difference of lengths do not decay on a finite timescale. This is consistent with previous results from stochastic simulations that the model with constant disassembly in which all biomolecules are shared does not yield simultaneous length control (*Mohapatra et al., 2017*). Noise must be included to observe this result; if fluctuations are not included, as in the deterministic ODE,

any initial state with the correct sum in lengths appears stable. This is because the zero eigenvalue causes such states to be marginally stable.

## Acknowledgements

We acknowledge useful discussions with Wallace Marshall, William Ludington, Shashank Shekhar, and Shane Mcinally. We thank Jie Lin and Ethan Levien for reading a draft of this manuscript and offering helpful feedback. We acknowledge support from National Science Foundation grants DMS-1502851 (TGF), grant PHY-1806818 (PK), grant DMR-1610737 (JK), MRSEC-1420382 (JK), the Simons Foundation (JK, LM), the A P Sloan Foundation (AA), NSF CAREER award 1752024 (AA), and the Kavli Institute (AA).

## Additional information

### Funding

| Funder | Grant reference number | Author |
|---|---|---|
| National Science Foundation | CAREER 1752024 | Ariel Amir |
| National Science Foundation | DMS-1502851 | Thomas G Fai |
| National Science Foundation | DMR-1610737 | Jane Kondev |
| National Science Foundation | MRSEC-1420382 | Jane Kondev |
| National Science Foundation | PHY-1806818 | Prathitha Kar |
| Alfred P. Sloan Foundation | | Ariel Amir |
| Kavli Foundation | | Ariel Amir |
| Simons Foundation | | Lishibanya Mohapatra Jane Kondev |

The funders had no role in study design, data collection and interpretation, or the decision to submit the work for publication.

### Author contributions

Thomas G Fai, Lishibanya Mohapatra, Jane Kondev, Ariel Amir, Conceptualization, Formal analysis, Writing—original draft, Writing—review and editing; Prathitha Kar, Formal analysis, Validation, Investigation, Visualization

### Author ORCIDs

Thomas G Fai ⓘ https://orcid.org/0000-0003-0383-5217
Prathitha Kar ⓘ https://orcid.org/0000-0002-4091-6860
Jane Kondev ⓘ https://orcid.org/0000-0001-7522-7144
Ariel Amir ⓘ https://orcid.org/0000-0003-2611-0139

### Decision letter and Author response

Decision letter https://doi.org/10.7554/eLife.42599.sa1
Author response https://doi.org/10.7554/eLife.42599.sa2

## Additional files

### Supplementary files

• Transparent reporting form

### Data availability

All data analyzed during this study are contained in the published studies cited in the references. Source code of the simulations used in our work can be found here: https://github.com/pkar96/

Agent-based-simulation (copy archived at https://github.com/elifesciences-publications/Agent-based-simulation).

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

## Appendix 1

### Stochastic simulation

The simulations are done using an agent-based model in which individual motors are tracked at each point of time, similar to that used in *Hendel et al. (2018)*.

The simulations are run for a fixed amount of time discretized into small time steps of size $\Delta t$. Parameters such as the diffusion constant $D$, velocity $v$, disassembly rate $d$, and injection rate $k_{on}$ are fixed. Four different variables are tracked for each motor:

- A variable indicating whether motors are located in the basal pool (value 0) or the flagellum (value 1). In the case of multiple flagella with shared motors, each flagellum is assigned a different positive integer.
- A Boolean variable indicating whether motors are undergoing ballistic or diffusive motion.
- The position $x_j(t)$ of motor $j$ for $j = 1, \ldots, M$.
- The amount of tubulin $t_j(t)$ carried by the motor ($t_j = \gamma T_f$ at the time of injection).

The following processes take place at each time step:

- The length of each flagellum $L_i(t)$ for $i = 1, 2$ is decreased by a constant amount $d\Delta t$ (in the case of constant disassembly) or by an amount $(d_0 + d_1 c_d(L))\Delta t$ (in the case of concentration-dependent disassembly), where the concentration $c_d(L)$ is computed by counting the number of diffusing motors within 1 μm of the flagellar tip.
- Motors are injected into the flagella with a probability $k_{on}M_f\Delta t$, where $M_f$ is the number of motors in the basal pool.
- Motors undergoing ballistic motion are advanced along the flagellum by a constant amount $v\Delta t$.
- Motors undergoing diffusive motion are advanced by a random amount drawn from a normal distribution with mean 0 and variance $2D\Delta t$.
- If a motor undergoing ballistic motion reaches the tip of the flagellum, the flagellar length is increased by the amount of tubulin $t_j$ carried by the motor. The motion of the motor is changed to diffusive and its position is reset to $L(t)$, that is the location of the flagellar tip.
- If the position of a motor moving diffusively exceeds the length of flagella, that is $x_j(t)>L(t)$, it is reflected according to $x_j(t) \rightarrow L(t) - (x_j(t) - L(t))$.
- If a diffusive motor reaches the base of the flagellum, it is taken up into the basal pool and stops diffusing. This enforces the absorbing boundary condition $c_d(0) = 0$.

By advancing the above processes in time, we obtain the length profiles $L_i(t)$.

To simulate the severing experiments, the length of one flagellum is shortened at a particular time and the motors and tubulin within the severed part of the flagellum are lost.

For those simulations with no protein replenishment (*Figure 3—videos 1*, *2*, *3* and *Figure 4—video 1*) we fix the total amount of tubulin $T$ and the total number of motors $M$. For those simulations in which we incorporate the replenishment of proteins in the basal pool (*Figure 3—videos 4*, *5*, *6* and *Figure 4—video 2*), the target protein numbers $\overline{T}_f$ and $\overline{M}_f$ are fixed along with the timescale of replenishment $\tau_r$. At each time step, there is a possibility of adding or removing a protein to or from the basal pool. The probability of addition or removal is proportional to the difference between the target number and the number of proteins currently in the basal pool.

The videos were made using the following parameters: $D = 1.7\,\mu\text{m}^2/\text{s}$, $k_{on} = 0.075\,s^{-1}$, $T = 38\,\mu\text{m}/\text{flagellum}$, $M = 200/\text{flagellum}$, $\gamma = 2.5 \times 10^{-4}$, $d_0 = 0.01\,\mu\text{m}/\text{s}$, $d_1 = 1.7 \times 10^{-3}\,\mu\text{m}^2/\text{s}$, $v = 2.5\,\mu\text{m}/\text{s}$. In *Figure 3—videos 4–6*, the replenishment timescale is $\tau_r = 300\,\text{s}$.

## Appendix 2

### Parameter estimation

To estimate the model parameters, we fit to experimentally-measured data. First, we extrapolate from *Marshall et al. (2005)* to estimate that the growth rate upon severing a flagellum is approximately 0.4 µm/min. This implies

$$\gamma k_{\mathrm{on}}MT - d \approx 0.4. \tag{64}$$

We next estimate the product $\gamma k_{\mathrm{on}}M$. To do so, we use the measured disassembly speed $d = 0.5\,\mu\mathrm{m/min}$ (*Marshall and Rosenbaum, 2001*) and an estimated initial tubulin pool of $T = 25 - 40\,\mu\mathrm{m}$. This yields $\gamma k_{\mathrm{on}}M = 2.3 \times 10^{-2}$–$3.6 \times 10^{-2}\,\mathrm{min}^{-1}$. The estimate $T$ for the tubulin pool of a single flagellum is based on the reported amount 76–94 $\mu$m for the total tubulin shared by two flagella (*Marshall et al., 2005*), allowing for some tubulin loss after severing. The estimate for $\gamma k_{\mathrm{on}}M$ is not expected to be very precise, particularly given that the total tubulin pool size from *Marshall et al. (2005)* was itself obtained by fitting the parameters of a related model to data on mutants with extra flagella.

We subsequently estimate $k_{\mathrm{on}}$ by fitting it to the steady-state length observed in the severing experiment. Returning to *Equation 35*, we may express $k_{\mathrm{on}}$ in terms of quantities that have already been measured or estimated:

$$k_{\mathrm{on}} \approx \left( \frac{L_{ss}}{v} + \frac{L_{ss}^2}{2D} \right)^{-1} \left( \frac{\gamma kM}{d}(T - L_{ss}) - 1 \right). \tag{65}$$

Plugging in the values for $\gamma kM, d$, and $T$ above as well as the measured values $v = 150\,\mu\mathrm{m/min}$ (*Kozminski et al., 1993*; *Buisson et al., 2013*), $D = 102\,\mu\mathrm{m}^2/\mathrm{min}$ (*Chien et al., 2017*), and $L_{ss} = 10\,\mu\mathrm{m}$ measured in *Ludington et al. (2012)*, we obtain $k_{\mathrm{on}} = 0.8 - 4.5\,\mathrm{min}^{-1}$. See the parameter values and definitions in *Table 1*.

### Boundary conditions

Here we discuss potential boundary conditions besides the diffusive sink $c_d(0) = 0$ described in Materials and methods. Repeating the previous calculation for a general prescribed concentration $c_d(0) = c_0$ at the flagellar base yields

$$J = \frac{k_{\mathrm{on}}(M - c_0 L)}{1 + k_{\mathrm{on}}L/v + k_{\mathrm{on}}L^2/2D}. \tag{66}$$

Therefore, to first order in $\Delta L = L - L_{ss}$ the effect of nonzero $c_0$ is equivalent to rescaling $M$ in *Equation 4*.

We also consider the scenario in which the basal pool concentration depends on the number of free molecular motors $M_f$ through $c_d(0) = AM_f/V$, where $A$ and $V$ are geometric parameters representing the flagellar cross sectional area $A$ and the basal pool volume $V$, respectively. Mass action kinetics of injection $J = k_{\mathrm{on}}M_f$ may be used to rewrite the basal concentration as $c_d(0) = AJ/(k_{\mathrm{on}}V)$. Substituting this expression into *Equation 66* results in

$$J = \frac{k_{\mathrm{on}}M}{1 + (k_{\mathrm{on}}/v + A/V)L + k_{\mathrm{on}}L^2/2D}. \tag{67}$$

Therefore, to first order in $\Delta L$ the effect of the boundary condition in this case is equivalent to rescaling $v \to v/(1 + (A/V)(v/k_{\mathrm{on}}))$ in *Equation 4*.

## Appendix 3

### Concentration-dependent disassembly model

#### Separate pools

We test various modes of coupling within the concentration-dependent depolymerization model and find that only the case of fully shared biomolecule pools is consistent with data (**Appendix 3—figure 1**). Unlike the constant disassembly models we have considered, there is some degree of length recovery after severing when only one biomolecule is shared. However, there is still no length equalization after severing because of the asymmetrical depletion caused by separate pools.

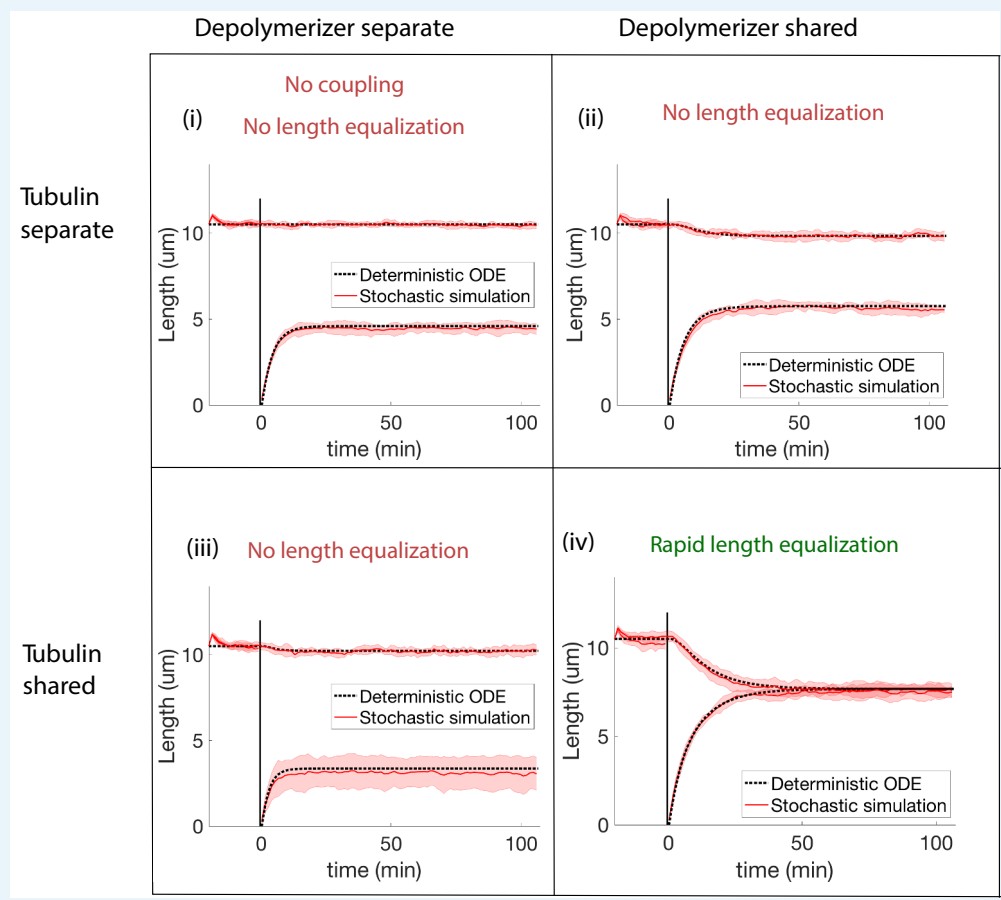

**Appendix 3—figure 1.** Concentration-dependent disassembly model: simulations of severing experiment with different modes of coupling between basal pools (and no replenishment of protein levels). We conclude that biomolecule pools are fully shared after ruling out all models that disagree with the rapid length equalization that occurs in severing experiments. (i) In the case of separate pools of both tubulin and depolymerizers, the unsevered flagellum does not decrease in length. (ii) When only depolymerizers are shared, the flagellar lengths do not equalize after severing. (iii) When only tubulin is shared, the flagellar lengths do not equalize after severing. (iv) When both tubulin and depolymerizers are shared, the flagella lengths rapidly equalize after severing, as observed in experiments.

## Biomolecules in excess

Flagellar growth in *Chlamydomonas* is limited by the supply of proteins. As discussed in Results, the importance of depletion effects is evidenced by the shortening of the long flagellum in severing experiments and by the cyclohexamide experiments of *Rosenbaum et al. (1969)* that blocked new protein synthesis and resulted in shorter flagella. Within the proposed active disassembly model this limiting-pool mechanism may arise in three ways: (i) both tubulin and motors are limited, (ii) tubulin is limited and motors are in excess, or (iii) tubulin is in excess and motors are limited. In *Figure 4* we considered case (i) given by *Equations 20 and 21* in which both biomolecules are limited. Here for completeness we consider the other two cases as well. Case (ii) may immediately be ruled out: if the rate-limiting IFT protein were in excess, their injection rate would not depend on flagellar length as observed in experiments (*Dentler, 2005*). A limiting pool of IFT proteins is therefore necessary for agreement with the severing data. We further consider the following model in which tubulin is in excess:

$$\frac{dL_1}{dt} = \gamma J T - d_0 - d_1 \frac{J L_1}{D}, \tag{68}$$

$$\frac{dL_2}{dt} = \gamma J T - d_0 - d_1 \frac{J L_2}{D}, \tag{69}$$

where $T$ is constant since tubulin is in excess and the flux $J$ is given as before by

$$J = \frac{k_{\mathrm{on}} M/2}{1 + k_{\mathrm{on}}(L_1 + L_2)/2v + k_{\mathrm{on}}(L_1^2 + L_2^2)/4D}. \tag{70}$$

Simulations show that the model *Equations 68 and 69* is consistent with severing data (*Appendix 3—figure 2*). Therefore, a limiting pool of IFT proteins is necessary for agreement with data, whereas tubulin may either be limited or in excess.

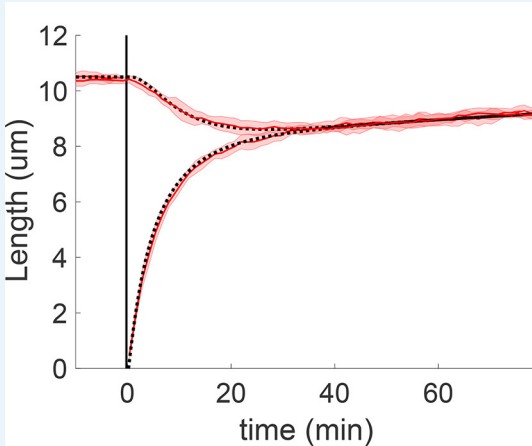

**Appendix 3—figure 2.** Concentration-dependent disassembly model *Equation 68 and 69* with tubulin in excess yields rapid length equalization consistent with severing experiments, here with replenishment timescale $\tau_r = 5 \, \mathrm{min}$.

## Linearization

Stability is established by linearizing about steady-state. We perform the linearization for the concentration-dependent disassembly model and discuss how the result generalizes to simultaneous length control with arbitrary flagellar number.

Let $\Delta L_1$ and $\Delta L_2$ be the deviations from steady-state such that $L_1 = L_{ss} + \Delta L_1$ and $L_2 = L_{ss} + \Delta L_2$. From (13), the flux $J$ is given by

$$J = \frac{k_{\text{on}}M/2}{1 + k_{\text{on}}((L_{ss} + \Delta L_1) + (L_{ss} + \Delta L_2))/2v + k_{\text{on}}((L_{ss} + \Delta L_1)^2 + (L_{ss} + \Delta L_2)^2)/4D}, \tag{71}$$

Keeping only terms up to first order in $\Delta L_1$ and $\Delta L_2$ in the denominator,

$$J = \frac{k_{\text{on}}M/2}{1 + k_{\text{on}}L_{ss}/v + k_{\text{on}}L_{ss}^2/2D + k_{\text{on}}(\Delta L_1 + \Delta L_2)/2v + k_{\text{on}}L_{ss}(\Delta L_1 + \Delta L_2)/2D} \\ + \mathcal{O}(\Delta L_1^2) + \mathcal{O}(\Delta L_1 \cdot \Delta L_2) + \mathcal{O}(\Delta L_2^2), \tag{72}$$

so that the first-order Taylor series expansion yields

$$J = \frac{k_{\text{on}}M/2}{1 + k_{\text{on}}L_{ss}/v + k_{\text{on}}L_{ss}^2/2D}\left(1 - \frac{k_{\text{on}}(\Delta L_1 + \Delta L_2)/2v + k_{\text{on}}L_{ss}(\Delta L_1 + \Delta L_2)/2D}{1 + k_{\text{on}}L_{ss}/v + k_{\text{on}}L_{ss}^2/2D}\right) \\ + \mathcal{O}(\Delta L_1 \cdot \Delta L_2) + \mathcal{O}(\Delta L_2^2), \tag{73}$$

Substituting this expression for the flux into the dynamical equations **Equations 20 and 21** results in

$$\frac{\mathrm{d}(\Delta L_1)}{\mathrm{d}t} = \frac{d_1 k_{\text{on}}M/2D}{1 + k_{\text{on}}L_{ss}/v + k_{\text{on}}L_{ss}^2/2D}(-\Delta L_1) \\ + \frac{\gamma k_{\text{on}}M/2}{1 + k_{\text{on}}L_{ss}/v + k_{\text{on}}L_{ss}^2/2D}\left(1 + \frac{(k_{\text{on}}/2v + k_{\text{on}}L_{ss}/2D)(T - (2 + d_1/\gamma D)L_{ss})}{1 + k_{\text{on}}L_{ss}/v + k_{\text{on}}L_{ss}^2/2D}\right)(-\Delta L_1 - \Delta L_2), \tag{74}$$

$$\frac{\mathrm{d}(\Delta L_2)}{\mathrm{d}t} = \frac{d_1 k_{\text{on}}M/2D}{1 + k_{\text{on}}L_{ss}/v + k_{\text{on}}L_{ss}^2/2D}(-\Delta L_2) \\ + \frac{\gamma k_{\text{on}}M/2}{1 + k_{\text{on}}L_{ss}/v + k_{\text{on}}L_{ss}^2/2D}\left(1 + \frac{(k_{\text{on}}/2v + k_{\text{on}}L_{ss}/2D)(T - (2 + d_1/\gamma D)L_{ss})}{1 + k_{\text{on}}L_{ss}/v + k_{\text{on}}L_{ss}^2/2D}\right)(-\Delta L_1 - \Delta L_2), \tag{75}$$

where we have retained terms up to first order in $\Delta L_1$ and $\Delta L_2$. Defining

$$a = \frac{\gamma k_{\text{on}}M/2}{1 + k_{\text{on}}L_{ss}/v + k_{\text{on}}L_{ss}^2/2D}\left(1 + \frac{(k_{\text{on}}/2v + k_{\text{on}}L_{ss}/2D)(T - (2 + d_1/\gamma D)L_{ss})}{1 + k_{\text{on}}L_{ss}/v + k_{\text{on}}L_{ss}^2/2D}\right),$$

$$b = \frac{d_1 k_{\text{on}}M/2D}{1 + k_{\text{on}}L_{ss}/v + k_{\text{on}}L_{ss}^2/2D},$$

we may write this system in the matrix form

$$\frac{\mathrm{d}}{\mathrm{d}t}\begin{pmatrix} \Delta L_1 \\ \Delta L_2 \end{pmatrix} = -\begin{pmatrix} a+b & a \\ a & a+b \end{pmatrix}\begin{pmatrix} \Delta L_1 \\ \Delta L_2 \end{pmatrix}. \tag{76}$$

The matrix above is the sum of a rank-one matrix and a diagonal perturbation. Roughly speaking, $a$ corresponds to the shared quantities whereas $b$ corresponds to the independent quantities. This system may be diagonalized in terms of the sum $\Sigma = \Delta L_1 + \Delta L_2$ and difference $\Gamma = \Delta L_1 - \Delta L_2$ to yield

$$\frac{\mathrm{d}}{\mathrm{d}t}\begin{pmatrix} \Sigma \\ \Gamma \end{pmatrix} = -\begin{pmatrix} 2a+b & 0 \\ 0 & b \end{pmatrix}\begin{pmatrix} \Sigma \\ \Gamma \end{pmatrix}. \tag{77}$$

Note that the eigenvalues

$$\lambda_\Sigma = -(2a+b), \tag{78}$$

$$\lambda_\Gamma = -b, \tag{79}$$

are both negative, so that the steady-state is stable, since $b$ is clearly positive and $T > 2L_{ss}$ implies the positivity of $a$. The fact that $\lambda_\Sigma$ and $\lambda_\Gamma$ are *distinct* is noteworthy as it provides a possible means to extract two independent parameters from experiment.

## Arbitrary flagellar number

This rank-one plus diagonal matrix structure also applies to the case of arbitrary flagellar number $N$ discussed in Results. Let $\Delta L_i := L_i - L_{ss}$ denote the deviation of the $ith$ flagellum from steady-state for $i = 1, \ldots, N$. The linearized equations satisfy

$$\Delta \mathbf{L} = M \Delta \mathbf{L}, \tag{80}$$

where $\Delta \mathbf{L} = (\Delta L_1, \ldots, \Delta L_N)$ and the matrix $M$ is of the form

$$M = -aR - bI, \tag{81}$$

with $R = \mathbf{1}\mathbf{1}^T$ the rank-one matrix satisfying $R_{ij} = 1$ for all $i, j$ and $I$ the $N \times N$ identity matrix, for example for N = 3

$$M = -a \begin{pmatrix} 1 & 1 & 1 \\ 1 & 1 & 1 \\ 1 & 1 & 1 \end{pmatrix} - b \begin{pmatrix} 1 & 0 & 0 \\ 0 & 1 & 0 \\ 0 & 0 & 1 \end{pmatrix} = \begin{pmatrix} -a-b & -a & -a \\ -a & -a-b & -a \\ -a & -a & -a-b \end{pmatrix}.$$

$M$ is straightforward to diagonalize. The vector $\mathbf{v}_1 := \mathbf{1} = (1, \ldots, 1)^T$ corresponding to the sum of all lengths is an eigenvector of $M$:

$$\begin{aligned} M\mathbf{1} &= -aR\mathbf{1} - bI\mathbf{1} \\ &= -(aN + b)\mathbf{1}. \end{aligned} \tag{82}$$

Further, for any vector $\mathbf{x} = (x_1, \ldots, x_N)$ such that $\sum_{i=1}^N x_i = 0$, we have $R\mathbf{x} = 0$ and

$$M\mathbf{x} = -b\mathbf{x}, \tag{83}$$

therefore it is an eigenvector with eigenvalue $\lambda = -b$. Note that the pairwise differences

$$\mathbf{v}_k = (0, \ldots 0, \underbrace{1}_{(k-1)^{st}\text{entry}}, \underbrace{-1}_{k^{th}\text{entry}}, 0, \ldots, 0)$$

for $k = 2, \ldots, N$ form a convenient basis for the space of such vectors $\mathbf{x}$ whose components sum to zero. In terms of the basis $\{\mathbf{v}_1, \mathbf{v}_2, \ldots, \mathbf{v}_N\}$ consisting of the sum and differences in lengths, the evolution equations diagonalize with eigenvalues

$$\lambda_\Sigma(N) = -a(N + b), \tag{84}$$

$$\lambda_\Gamma(N) = -b, \tag{85}$$

for the sum and differences, respectively, so that $\lambda_\Sigma(N)$ is an eigenvalue of multiplicity 1 and $\lambda_\Gamma(N)$ is an eigenvalue of multiplicity $N - 1$. Note that in addition to the explicit dependence of $\lambda_\Sigma(N)$ and $\lambda_\Gamma(N)$ on $N$ there is an implicit number-dependence through the steady-state length (and potentially the size of the pool $T$). Since $\lambda_\Sigma(N) \neq \lambda_\Gamma(N)$, the dynamics involve two distinct timescales $\tau_\Sigma(N) = \lambda_\Sigma(N)^{-1}$ and $\tau_\Gamma(N) = \lambda_\Gamma(N)^{-1}$ corresponding to these two eigenvalues.

## Different depolymerizer and rate-limiting IFT protein

In the description of the concentration-dependent disassembly model in Results, for convenience we made the assumption that the depolymerizer is the rate-limiting IFT protein. Here, we consider the more general case that the depolymerizer is not the rate-limiting IFT protein, but rather a different protein that is carried to the flagellar tip by IFT and present in excess in the basal pool. The formula derived in the manuscript for the flux $J$ represents the flux of IFT particles, and in this case the flux of depolymerizer is $KJ$, where the capacity $K$ of depolymerizers per IFT particle is assumed constant. Assuming the depolymerizer diffuses

back from the flagellar tip with diffusivity $D'$, the concentration $c'_d$ of diffusing depolymerizer proteins satisfies

$$D'\frac{\partial c'_d}{\partial x} = KJ,\qquad(86)$$

so that the concentration at position $x$ along the flagellum satisfies $c'_d(x) = c'_0 + \frac{KJ}{D'}x$ under the boundary condition $c'_d(0) = c'_0$. (Primes are used to distinguish the parameters for the depolymerizer from those of the rate-limiting IFT protein.) In this case the dynamical equations for length become

$$\frac{\mathrm{d}L_1}{\mathrm{d}t} = \gamma J(T - L_1 - L_2) - d_0 - d_1\left(c'_0 + \frac{KJL_1}{D'}\right),\qquad(87)$$

$$\frac{\mathrm{d}L_2}{\mathrm{d}t} = \gamma J(T - L_1 - L_2) - d_0 - d_1\left(c'_0 + \frac{KJL_2}{D'}\right),\qquad(88)$$

Therefore **Equations 20 and 21** derived in Results remain valid in this more general case upon making the identification $d_0 \rightarrow d_0 + d_1 c'_0$ and $d_1 \rightarrow \frac{KD}{D'}d_1$. We have verified these results through agent-based simulations (**Appendix 3—figure 3a**).

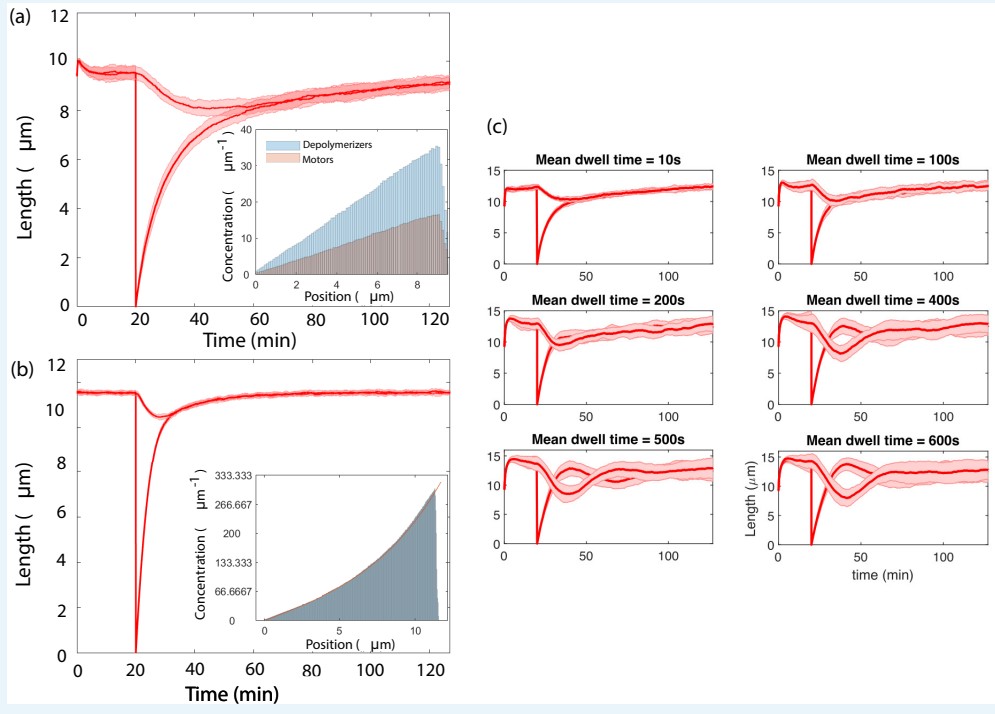

**Appendix 3—figure 3.** Simultaneous length control is achieved by different versions of the concentration-dependent disassembly model. (**a**) Results from agent-based simulations with different populations of depolymerizer and rate-limiting IFT protein, using capacity $K = 2$, (inset) Concentrations along the flagellum of diffusing IFT proteins and depolymerizers, (**b**) An exponential profile is generated by motile proteins that bind and unbind from the flagellum, move ballistically in the anterograde direction when bound, and undergo diffusion when unbound, (inset, fit to **Equation 89**), (**c**) Agent-based simulations of processive depolymerization, in which depolymerizers diffusing near the flagellar tip have some probability of binding and removing tubulin at a fixed rate. We perform simulations of severing over a range of mean depolymerization times from 10 s to 600 s.

## Exponential concentration gradient

To show that our model allows for non-linear concentration gradients, we set up agent based simulations which would lead to an exponential concentration distribution, similar to *Naoz et al. (2008)*.

In our simulations, after injection motors travel ballistically to the tip of the flagellum where they begin to diffuse back toward the base as discussed in Appendix 1. In addition motors may detach with rate $k_d$ to undergo diffusive motion and reattach with rate $k_a$ to switch from diffusive to ballistic motion. This leads to an exponential distribution with the concentration profile of diffusing motors being

$$C(x) = \frac{Jv}{\sqrt{d^2 k_d^2 + 4k_a v^2 d}}(\exp(\lambda_+ x) - \exp(\lambda_- x)), \tag{89}$$

where as before $J$ = injection rate = $k_{on} M_f / 2$ and $\lambda_{\pm}$ is given by

$$\lambda_{\pm} = \frac{-1 \pm \sqrt{1 + (4k_a/D)(v/k_d)^2}}{2v/k_d}. \tag{90}$$

The concentration profile of diffusing motors with a fit to the above equation is shown along with the length regulation of flagella in *Appendix 3—figure 3b* for rates $k_a = 6.3 \times 10^{-2}$ s$^{-1}$ and $k_d = 3.13 \times 10^{-4}$ s$^{-1}$. The results are found to be qualitatively similar to the case of a linear concentration gradient, supporting our claim that concentration-dependent disassembly is able to control lengths independent of the precise form of the concentration gradient.

## Processive depolymerizers

By taking the disassembly rate to depend on local depolymerizer concentration, we have implicitly assumed that the depolymerizer acts non-processively. However, this is not a fundamental restriction. We have used our agent-based model to explore a mechanism of processive depolymerization, in which depolymerizers diffusing within 1 micron of the flagellar tip bind at rate $9.4 \times 10^{-2}$ s$^{-1}$ and depolymerize at a rate of $9.7 \times 10^{-4}$ µm/s. The depolymerization duration is drawn from an exponential distribution with prescribed mean. We have verified that this model achieves simultaneous length control over a range of mean depolymerization times. In *Appendix 3—figure 3c* we show the results of simulations using mean depolymerization times from 10 s (corresponding to 0.01 µm) up to 10 min (corresponding to 0.6 µm). For mean depolymerization times up to 100 s, we find the results to be qualitatively similar to the non-processive case, whereas for longer mean depolymerization times the steady-state lengths exhibit oscillations about the steady-state. The emergence of oscillations is not entirely surprising since processive depolymerization effectively introduces into the equations a time delay, which is known to give rise to oscillations in many contexts (*Richard, 2003*).

