## [Decision Letter]

[Editors’ note: this article was originally rejected after discussions between the reviewers, but the authors were invited to resubmit after an appeal against the decision.]

Thank you for submitting your work entitled "Length regulation of multiple flagella that self-assemble from a shared pool of components" for consideration by *eLife*. Your article has been reviewed by three peer reviewers, one of whom is a member of our Board of Reviewing Editors, and the evaluation has been overseen by a Senior Editor. The reviewers have opted to remain anonymous.

Our decision has been reached after consultation between the reviewers. Based on these discussions and the individual reviews below, we regret to inform you that your work will not be considered further for publication in *eLife*.

The authors perform a theoretical study of the regulation flagellar length in *Chlamydomonas* based on intraflagellar transport of motors and tubulin subunits that govern assembly and disassembly. A key question is how the length of two flagella can be coordinated by exchange of components. This is an important question. The present paper extends an earlier model by Hendel, Thomson and Marshall. It presents simple analytic expressions for the length dynamics and steady state lengths in several scenarios of components that are shared or separate between cilia. After thorough discussion the reviewers concluded that this work is not a major advance as compared to the earlier work by Hendel et al. It provides an extension of that work which is of interest to specialists but does not represent a fundamental advance.

Reviewer #1:

The authors perform a theoretical study of the regulation flagellar length in *Chlamydomonas* based on intraflagellar transport of motors and tubulin subunits that govern assembly and disassembly. A key question is how the length of two flagella can be coordinated by exchange of components.

The paper is largely well written and interesting. First a reduced model is presented with several simplifying assumptions such as separation of timescales and simplified kinetic rules. In the context of this reduced model the coordination of two cilia is then discussed. It is shown that if a molecular component is not shared between two cilia then length can be coordinated. If all components are shared then coordination needs to be more subtle. The authors suggest that concentration dependent depolymerization could be responsible for length coordination.

This work is interesting but it also has shortcomings. Reading the Introduction, the paper makes a strong impression. However, when I worked through the main part of the paper weaknesses became apparent and the discussion seemed to be rather superficial. In the end it remains unclear what advance in our understanding the work actually achieves. In its present form I do not think that this work is suitable for publication in *eLife*.

Major points:

1) The main motivation of the paper is to provide a possible explanation of the cilia severing experiment shown in Figure 1B. This is an important and interesting problem. However, the manuscript fails to really advance this issue. The authors rule out the two independent flagella (Figure 3Bi) because they cannot account for the coordinated behavior of the experiment. It seems that the proposed model of coordinated flagella shown in Figure 4 does also not really capture the key feature of coupled flagella observed in experiments: the flagellum that is not severed shrinks to almost half its original length and then both flagella grow together to their final length. In fact, Figure 4 which is a key figure of the paper is not well presented. In a severing experiment the longer flagellum should start from the steady state length which both flagella reach at long times. Another problem with Figure 4 is that only stochastic simulations are shown. It would be better to show the true average which is obtained by solving the deterministic equations. Then it would be clear if the longer flagellum first reaches a minimum before it again increases its length to reach the steady state. The stochastic simulations can be misleading as the fluctuations conceal this important feature in the behavior or even give the impression of length minima but which arise only from the noise.

2) It is an interesting but not deep insight (given the simplifications of the model), that if all components are shared then only the total length is fixed but individual lengths are independent. As the authors show this feature is a result of the simplifications used. Taking more realistic aspects into account, such as the concentration dependent depolymerization, this degeneracy in the model is removed. However, there are most likely other possibilities that could provide such a lift of the degeneracy. The fact that one mechanism can lift the degeneracy is interesting but does not constitute a strong result given that the model fails to qualitatively account for the experimental data.

3) When discussing the cases where one component (M or T) is not shared but exists in two different pools, it is implicitly assumed that the separate pools (e.g. T_1_ and T_2_) are equal. However, this is not stated clearly and it is not clear why they should be equal if they are completely independent.

Reviewer #2:

The authors have analyzed a class of models for length control of the two cilia of *Chlamydomonas*. The interesting result is that a naive model in which both tubulin and motor pools are shared between the two cilia doesn't work: only the summed ciliary length is determined and the individual lengths are undetermined. To get length control for two cilia, the authors add a length-dependent depolymerization, which can be satisfied by a depolymerase whose concentration increases with length (which happens if anterograde transport of the depolymerase is advective but retrograde transport diffusive).

My main concern is: what is different/new compared to the Hendel et al. (2018) paper? Is this paper wrong, in the sense that their mechanism (which has no length-dependent depolymerase) does not work, contrary to their claims? If this is the case, then then this point must be stressed. If the Hendel et al. paper is correct, then more justification of the novelty of the current work is necessary.

The following points need to be addressed in the manuscript

1) What is the concentration profile of the depolymerase? Please add this in a figure. I assume that it is a linear increasing concentration from the base (as in the Hendel paper) but I would like to see this.

2) Under what conditions can depolymerases regulate a single cilium? Is a limiting tubulin pool necessary? Is a limiting tubulin pool necessary for the case of two cilia? Is a limiting pool of IFT motors necessary? The general point I am raising is that length-dependent depolymerization is a strong assumption and perhaps it is sufficient under very broad conditions. What are they? Then this point needs to be discussed.

3) The stochastic aspect of the work is irrelevant as it is not used and should be removed from the paper (perhaps put into another paper).

Reviewer #3:

The paper describes a very interesting study of the possible models that can account for the simultaneous length control in the flagella of a single-cell organism. The study is illuminating, and shows how simple models can help to shed light on the microscopic processes, simply from the analysis of large-scale dynamics. I have a few comments:

1) How many MT are in each flagellum? How is this number controlled? Is this also a dynamic variable that self-organizes? The implicit assumption is that this is a constant (determined at the base, and therefore not dependent on length?), but should be explained.

2) The authors consider that the ballistic motion of the motors to the growing tip is uninterrupted, is motors and their cargo do not detach from each other or from the MT track until the tip. Is this known to be a good approximation of this system? It certainly simplifies the analysis, as they do not need to solve the spatial distribution of the density of motors, and cargo, along the flagella's length.

I would suggest discussing how this is different from models of the growth and steady-state of actin-based protrusions, where dis-assembly is at the base, and the length is a result of force balance (see Naoz et al., 2008; Orly et al., 2014).

3) The question about the return current of tip-directed motors is also an open issue for myosin motors along actin-filled protrusions. Interesting dynamics and traffic jams have been observed and modeled (see Yochelis et al., 2015; Pinkoviezky and Gov, 2014, 2017), I suggest contrasting with the situation in the flagella.

4) The noise in the reactions that control the growth at the tip should give rise to a term in (1) that has noise multiplied by J and T_f_? Would this change the dynamics?

5) MTs undergo catastrophes and recoveries, and the data traces seem to show this. Does this type of dynamics matter for the model? This should be discussed.

[Editors’ note: what now follows is the decision letter after the authors submitted for further consideration.]

Thank you for resubmitting your work entitled "Length regulation of multiple flagella that self-assemble from a shared pool of components" for further consideration at *eLife*. Your revised article has been favorably evaluated by Naama Barkai (Senior Editor), a Reviewing Editor, and two reviewers.

The reviewers have discussed the reviews with one another and the Reviewing Editor has drafted this decision to help you prepare a revised submission.

Summary:

The manuscript discusses the length regulation of the pair of flagellae of the green algae *Chlamydomonas*. The paper builds on earlier work of the Wallace Marshall group which proposed basic concepts of length regulation via tubulin transport along flagellae controlling assembly together with disassembly. Coupling of flagellae can be mediated by shaped molecular pools.

The present paper provides two important results:

i) Detailed theoretical analysis of the shared limiting-pool mechanism, and showing that by itself this mechanism does not lead to length equilibration of the two flagella.

ii) A careful study of a length regulation by length dependent depolymerization, showing that length dependent depolymeriyation together with shared molecular pools can account for the experimental observations.

However, the paper also has some serious weaknesses. In particular:

1) The experimental basis for their proposal of length dependent depolymerization are much weaker than claimed. In fact, Piao et al., 2009 and 2013, do not give evidence for depolymerases at the flagellar tip in steady state.

2) The claim that length dependent depolymerization is the only mechanism to account for the data is an overstatement.

3) The discussion of previous work (Hendel et al.) that did find length regulation as seen in experiments without the need of length dependent depolymerization is discussed only superficially and with a somewhat negative attitude. This previous work should be taken more seriously and discussed more carefully. It does show that length regulation does not require length dependent depolymerization. However, this seems to be resulting from different details and assumptions in the model.

The authors should tone down their claims and more carefully relate their results to previous work. Then this paper could become a very valuable contribution that clarifies subtle but general aspects of length regulation and provides novel insights in the possible role of length dependent depolymerization.

The authors have to revise their manuscript carefully before it could be suitable for publication in *eLife*.

Essential revisions:

– The authors discuss a mechanism for length dependent depolymerization. However, several arguments are unclear. The authors begin with the assumption in the Introduction: "We will assume for now that the rate-limiting protein is kinesin-2, which is the molecular motor responsible for transport toward the tip of the flagellum". But, then in subsection “Tubulin shared, motors shared and concentration-dependent disassembly” the authors write "although to this point we have assumed the rate-limiting protein is kinesin-2, in fact the model is valid for any rate-limiting IFT protein. Therefore, in what follows we assume that the rate-limiting protein is a depolymerizer having the same motion as kinesin-2, uninterrupted ballistic motion to the tip followed by diffusive motion to the base…"

The authors should clarify if there is a family of kinesins that possess both these properties, namely, anterograde transport of tubulin dimers in a directed manner along a microtubule as well as microtubule depolymerase activity. It seems no such motor is known.

– In support of their claim, the authors quote the experimental observations of Piao et al. Piao et al. clearly state in their paper that kinesin-13 "was transported by IFT into flagella during flagellar shortening." Further elaborating on this mechanism, Wang et al. (2013) reported that "CrKin13 was barely detectable in the flagella of steady state cells, as shown previously (Piao et al., 2009). However, during rigorous phase of flagellar assembly at 15 and 30 min after deflagellation, CrKin13 was found to be enriched in the flagella and decreased to normal level in fully assembled flagella". Thus, unlike the mechanism proposed by Fai et al. in this paper, experiments indicate very little presence of kinesin-13 depolymerases in the flagella under normal circumstances. Only amputation of one of these trigger a signal that leads to rapid entry of the kinesin-13 into the uncut flagellum for its depolymerization that supplies tubulins for the initial regeneration of the amputated flagellum. Thus, the roles of kinesin-2 and kinesin-13 and their presence or absence in the flagellum should be treated separately and cannot be represented by an all encompassing single motor. The relation to experiments and the required properties of motors should be discussed more carefully (see also next point)).

– Piao et al. demonstrated the depolymerase activity of kinesin-13 family (and not kinesin-8 family) in the microtubule disassembly in cilia. Kinesin-13 diffuses along microtubules to target either end and are not known to transport tubulin. Therefore, the claim of experimental support of the model seems to be a too strong statement.

– Subsection “Tubulin shared, motors shared and concentration-dependent disassembly”, the authors state "Unlike constant disassembly models in which a limiting-pool mechanism is essential for length control,…" This appears to be an incorrect statement. Some constant-disassembly models, e.g., Time-of-flight (TOF) model, which also assumes constant disassembly, does not assume "limiting pool". Instead TOF assumed "differential loading" of IFT particles, as indicated by the experiments of Wren et al.

Therefore, the statement "As we shall see, the constant disassembly models do not result in the rapid length equalization observed experimentally" is an overstatement and oversimplification.

– The key assumptions of Fai et al.'s model seem to be the following: (a) a concentration gradient of the depolymerases along the length of the flagellum and (b) the local depolymerization rate is proportional to the local concentration of the depolymerases. There appears to be an additional implicit assumption: the depolymerase is non-processive in its depolymerase activity (not to be confused with processivity in motility). Otherwise, depolymerases loaded on to the MT plus-end at a location closer to the ciliary tip may continue to depolymerize even when the MT becomes shorter and its plus-end reaches points where the depolymerase concentrations are quite different.

– The possibility of a concentration gradient of the motors arose naturally already in the paper of Hendel et al. (see page 667 of Hendel et al.). However, Hendel et al. did not associate this concentration gradient with that of depolymerases, which seems to be the appropriate picture.

– The authors present "testable predictions". However there are some problems. The experimental results already reported by Piao et al., 2009 and 2013, do not provide evidence for the presence of depolymerases at the tip in steady state. Also based on these experiments the existence of a concentration gradient of the depolymerases in the steady-state (that too, an increasing concentration towards the tip) seems to be unlikely.

The second testable prediction that the authors mention would not be a clear test for validation/refutation of their model. For all those models that depend on a dynamic balance of the rates of polymerization and depolymerization in the steady-state of the flagellum, lowering the entry flux of the IFT particles would lead to overall depolymerization although the local rate of depolymerization of the microtubules at the plus end would remain unchanged.

– The slight difference in the expressions for L_ss_ derived by Hendel et al. and that of Fai et al. arises from difference in the scenarios considered by the two. The two assumptions made by Hendel et al. were clearly stated in their paper. The first assumption of "a constant source of free motor protein at the tip" and the second assumption of "the approximation in which motors that have reached the base immediately transport back to the tip" are exactly the two conditions ("no tubulin depletion" and "instantaneous ballistic motion") that Fai et al. point out in reducing their result to that of Hendel et al. In this sense their derivation is only slightly improved compared to that of Hendel et al.

– I find the claims of the paper that Hendel’s model is somehow incorrect and that the present manuscript solves all open issues unconvincing. Hendel et al. show a scenario where length control works, probably based on different detailed assumptions. This should be more clearly discussed and not superficially mentioned as in the paragraph three of the Discussion.

– The authors discuss the case where molecular components stabilize their levels (Equations 18 and 19) and point out the problem that in steady state such a relaxation to fixed levels breaks length regulation. It remains unclear if that is also a problem for the proposed model based on length dependent depolymerization. Also, it would be important to know what happens when the motors alone are relaxing to a fixed level but the tubulin is limited and fixed. Would that be similar to the results of Hendel et al.?

– The authors seem to miss relevant references: Varga et al. (2009); Klein et al. (2005); Johann et al. (2012).‏ These should be cited and their consequences for this paper should be discussed.

---

## [Author Response]

[Editors’ note: this article was originally rejected after discussions between the reviewers, but the authors were invited to resubmit after an appeal against the decision.]

The authors perform a theoretical study of the regulation flagellar length in Chlamydomonas based on intraflagellar transport of motors and tubulin subunits that govern assembly and disassembly. A key question is how the length of two flagella can be coordinated by exchange of components. This is an important question. The present paper extends an earlier model by Hendel, Thomson and Marshall. It presents simple analytic expressions for the length dynamics and steady state lengths in several scenarios of components that are shared or separate between cilia. After thorough discussion the reviewers concluded that this work is not a major advance as compared to the earlier work by Hendel et al. It provides an extension of that work which is of interest to specialists but does not represent a fundamental advance.

We appreciate the detailed and useful comments from the three reviewers. In light of these comments, we have performed full stochastic simulations to verify our analytical results and clarify their relation to the model described in Hendel et al. This has led to a new and surprising result that we are very excited about: of the five models discussed in our paper, only one leads to length control consistent with the available experimental data, and it is not the one of Hendel et al. Instead, it is the model in which length control is achieved by length-dependent depolymerization. The essence of these new results are briefly explained below. We have also found supporting experimental evidence for this model in published papers on *Chlamydomonas reinhardtii* (Piao et al., 2009) and on *Leishmania major* (Blaineau et al., Current Biology 17, 778–782 (2007)), which both describe the role of a microtubule depolymerizing protein (kinesin 13) in the assembly and length regulation of flagella. With these new results in hand, we feel that our theoretical work significantly advances our understanding of flagellar length control. It challenges the existing paradigm of control by length-dependent polymerization by switching focus on length-dependent depolymerization. Finally, our model provides concrete predictions for new experiments.

We would like to emphasize that these results allow us to definitively address a major concern of the reviewers, which is how our manuscript advances the field beyond the work of Hendel et al. We have used new agent-based simulations to reproduce the results of Hendel et al. and confirmed that their model involves an additional control mechanism of the two motor pools associated with each flagellum; in their simulations of the”severing experiment” (in which one of the two flagella is cut), the amount of IFT lost upon cutting is added (in silico) to the basal pool. This assumption is required for agreement with the experiments. In effect, regeneration of the flagella to their original length after one has been severed is achieved in the model through this control mechanism that effectively monitors the total number of motors associated with each flagellum. There is no biological justification for this assumption, and it seems highly implausible as it would require the cell to count both the freely diffusing motors and those taking part in transport by IFT-particles, and to maintain this number at a fixed value. Our new results show that removing this additional control mechanism, while leading to length control of a single flagellum (as described by Hendel et al.), is incapable of regenerating equal lengths of both flagella after one is cut, in clear contradiction with experiment. This subtlety of the Hendel et al. model was missed by us as well, leading us to conclude previously that their model is consistent with experiments. We only became aware of the disparity upon careful and detailed stochastic simulations in which we attempted to replicate the work of Hendel et al. To our surprise, this left us with the conclusion that all available experimental evidence now clearly points in the direction of flagellar length being controlled by length-dependent depolymerization (the last model discussed in our manuscript). As stated above, this is a paradigm shifting proposal in this field and we can offer concrete predictions for experiments that will test it (some of which have already been done in published papers mentioned above). The following is a list of the main changes, with references to relevant videos from full stochastic simulations included in the new submission

1) The Langevin equation (stochastic differential equation) formulation has been removed and replaced by a deterministic ordinary differential equation throughout the manuscript. We have included new stochastic simulations (see the new Appendix 1 and associated videos) that model the assembly process with more detail than the differential equation model and include stochastic effects.

2) Figure 2 describing the length dynamics of a single flagellum has been remade to emphasize the emergence of length-dependent injection.

3) Figure 3 has been remade without assuming instantaneous recovery of the original number of motors in the flagellum (see also the new Figure 3—videos 1–3). These results rule out all four versions of the constant disassembly model.

4) We have considered a model of protein replenishment based on external control of basal body concentrations (Equations 18 and 19). Somewhat counterintuitively, this further destabilizes the constant disassembly models, as we show by an analytic argument included in our response and subsection “Controlling protein levels in the basal pool is incompatible with constant disassembly” of the revision and illustrate in the new Figure 3—videos 4–6.

5) Figure 4 has been remade and merged with the previous Figure 6. In Figure 4C we have added quantitative fits of the concentration-dependent disassembly model to severing data.

6) We have considered variants of the active disassembly model in which not all biomolecules are shared. Our results show that, in addition to active disassembly, all biomolecules must be shared to obtain agreement with the severing experiments (see the new Appendix 3—figure 1). We have also shown that, whereas a limiting pool of IFT proteins is required to capture the experimentally-observed injection rates, tubulin may be in excess (new Appendix 3—figure 2).

Reviewer #1:The authors perform a theoretical study of the regulation flagellar length in Chlamydomonas based on intraflagellar transport of motors and tubulin subunits that govern assembly and disassembly. A key question is how the length of two flagella can be coordinated by exchange of components.The paper is largely well written and interesting. First a reduced model is presented with several simplifying assumptions such as separation of timescales and simplified kinetic rules. In the context of this reduced model the coordination of two cilia is then discussed. It is shown that if a molecular component is not shared between two cilia then length can be coordinated. If all components are shared then coordination needs to be more subtle. The authors suggest that concentration dependent depolymerization could be responsible for length coordination.This work is interesting but it also has shortcomings. Reading the Introduction, the paper makes a strong impression. However, when I worked through the main part of the paper weaknesses became apparent and the discussion seemed to be rather superficial. In the end it remains unclear what advance in our understanding the work actually achieves. In its present form I do not think that this work is suitable for publication in eLife.

We thank the reviewer for the careful reading of the manuscript and helpful feedback. Inspired by the reviewers’ comments, we have made substantial changes that we believe address the shortcomings in our original submission and significantly improve this work.

As mentioned above, we have used new agent-based simulations to reproduce the results of Hendel et al. and confirmed that their model involves an additional control mechanism of the two motor pools associated with each flagellum; in their simulations of the”severing experiment” (in which one of the two flagella is cut), the amount of IFT motors lost upon cutting is added (in silico) to the basal pool. As we show in the Author response image 1 (to be compared with Figure 6A from Hendel et al.), this assumption is required for agreement with the experiments. In effect, regeneration of the flagella to their original length after one has been severed is achieved in the model through this control mechanism that effectively monitors the total number of motors associated with each flagellum. There is no biological justification for this assumption, and it seems highly implausible as it would require the cell to count both the freely diffusing motors and those taking part in transport by IFT-particles, and to maintain this number at a fixed value. Our new results show that removing this additional control mechanism, while leading to length control of a single flagellum (as described by Hendel et al.), is incapable of regenerating equal lengths of both flagella after one is cut, in clear contradiction with experiment.

The detailed stochastic simulations performed for the new submission have led us to realize a subtlety of the Hendel et al. model of severing experiments: upon correctly accounting not only for the tubulin, but also for the motors lost upon severing, all variants of the constant disassembly model become inconsistent with the data. Our work explains why length-dependent disassembly, not length-dependent assembly, is the essential ingredient for length control in *Chlamydomonas*.

In our model this length-dependent disassembly is achieved by ballistic-to-diffusive IFT motion of a depolymerizer, which results in a concentration gradient in steady-state. We have also found supporting experimental evidence for this model in published papers on *Chlamydomonas reinhardtii* (Piao et al., 2009) and on *Leishmania major* (Blaineau et al., Current Biology 17, 778–782 (2007)), which both describe the role of a microtubule depolymerizing protein (kinesin 13) in the assembly and length regulation of flagella.

Our revised submission has three main results, as described in detail later on in the response. The first result is that models in which only one protein is separate do not capture the rapid length equalization observed experimentally. This is because of the asymmetry in protein levels after severing, an effect that was initially missed by us as well as Hendel et al., and it allows us to rule out all of the constant disassembly models on a case-by-case basis via their short-time behavior.

Whereas previously we focused on the short-time dynamics and ignored replenishment of proteins, on longer-timescales the importance in *Chlamydomonas* of replenishing pool levels was shown by experiments that used cyclohexamide to block new protein synthesis after severing and resulted in shorter flagella (Rosenbaum et al., 1969), an effect that others had included previously in theoretical models including Hendel et al. Our second result is that protein replenishment is fundamentally inconsistent with constant disassembly, as shown by a simple analytic argument. This allows us to rule out constant disassembly models on the basis of the long-time behavior over which protein replenishment takes place.

Our third main result is that the active disassembly model we propose is consistent with both the rapid length equalization and the protein replenishment observed in severing experiments; we show that the model agrees with data from Ludington et al. (2012) and Rosenbaum et al. (1969). With these new results in hand, we feel that our theoretical work significantly advances our understanding of flagellar length control. Finally, our model provides concrete predictions for new experiments which we can now propose.

Major points:1) The main motivation of the paper is to provide a possible explanation of the cilia severing experiment shown in Figure 1B. This is an important and interesting problem. However, the manuscript fails to really advance this issue. The authors rule out the two independent flagella (Figure 3Bi) because they cannot account for the coordinated behavior of the experiment. It seems that the proposed model of coordinated flagella shown in Figure 4 does also not really capture the key feature of coupled flagella observed in experiments: the flagellum that is not severed shrinks to almost half its original length and then both flagella grow together to their final length. In fact, Figure 4 which is a key figure of the paper is not well presented. In a severing experiment the longer flagellum should start from the steady state length which both flagella reach at long times. Another problem with Figure 4 is that only stochastic simulations are shown. It would be better to show the true average which is obtained by solving the deterministic equations. Then it would be clear if the longer flagellum first reaches a minimum before it again increases its length to reach the steady state. The stochastic simulations can be misleading as the fluctuations conceal this important feature in the behavior or even give the impression of length minima but which arise only from the noise.

We have revised Figure 4 so that it now includes all relevant aspects of the severing experiment. We have redone all figures to show that the flagellar lengths are in steady-state prior to severing. In Figure 4B we focus on the short-time dynamics without protein replenishment and illustrate the rapid length equalization exhibited by the model. In Figure 4C we add protein replenishment as previously mentioned and show that, when the model is fit to data, it is able to capture both the short-time behavior and the slow recovery to the original steady-state.

We have also taken to heart the reviewer’s suggestion to solve the deterministic ODE rather than showing realizations of the stochastic differential equation (SDE). We have removed the SDE formulation throughout the manuscript, and now include results from the deterministic ODE as well as new agent-based stochastic simulations.

2) It is an interesting but not deep insight (given the simplifications of the model), that if all components are shared then only the total length is fixed but individual lengths are independent. As the authors show this feature is a result of the simplifications used. Taking more realistic aspects into account, such as the concentration dependent depolymerization, this degeneracy in the model is removed. However, there are most likely other possibilities that could provide such a lift of the degeneracy. The fact that one mechanism can lift the degeneracy is interesting but does not constitute a strong result given that the model fails to qualitatively account for the experimental data.

As shown in the new Figure 4C, we are now able to show that fitting the concentration-dependent disassembly model to data yields agreement with the primary qualitative features of severing experiments.

Although as the reviewer suggests that the degeneracy of the constant disassembly model could be lifted in multiple ways, we believe there are several compelling arguments for why concentration-dependent depolymerization is the leading candidate. First, there is experimental evidence in support of our model: it is known that in the absence of IFT, depolymerization is 50-fold slower [1], and a particular IFT protein, kinesin-13, has been shown to act as depolymerizers in both *Chlamydomonas* [2] and *Leishmania* [3]. Second, from a theoretical perspective this model fits well with all phases of the severing experiment, in particular the length equalization after severing.

After the first stage of rapid length equalization, protein replenishment is needed for the recovery of the flagellar lengths back to their original steady-state as demonstrated by experiments of Rosenbaum et al. that use cyclohexamide to block new protein synthesis after severing and resulted in shorter flagella [4]. Protein replenishment was included in the Hendel et al. model as well. Importantly, we have now realized that replenishing proteins based on protein levels in the basal body implies that length control *must* include non-constant disassembly, in contrast to the prevailing model. This can be shown by the following simple argument: the final phase of severing experiments implies a replenishment of the pools at some relatively slow timescale *τ_r_*. Assuming that the cell maintains target protein concentrations in the basal body, in steady-state these concentrations will be equal to their target levels. However, in the constant disassembly model the growth rate is determined solely by basal body concentrations, so that length drops out of the steady-state equations. That is, the constant disassembly is inconsistent not only with the first phase of rapid length equalization, but also with a plausible control mechanism describing the second phase of slow recovery.

More precisely, to include external control with a target tubulin level T¯f and target motor number M¯f in the basal body, we include the following equations:τrdTdt=T¯f−TfτrdMdt=M¯f−Mf.

In steady-state, Tf=T¯fand Mf=M¯fso that in steady-state the assembly term γkonM¯fT¯f is constant, and the only way to obtain a balance point is through length-dependent disassembly. Note that, although here we have used a linear restoring force for simplicity, the conclusion also holds for general nonlinear restoring forces having a stable steady-state at Tf=T¯f and Tm=M¯f.

We have included this argument in the revision.

3) When discussing the cases where one component (M or T) is not shared but exists in two different pools, it is implicitly assumed that the separate pools (e.g. T_1_ and T_2_) are equal. However, this is not stated clearly and it is not clear why they should be equal if they are completely independent.

We thank the reviewer for this important comment. Indeed, in the case of proteins with separate pools, after severing the amounts in each pool should be different. This point was neglected by us in our original submission and by Hendel et al. Upon properly accounting for this asymmetry in protein levels after severing, we find that the constant disassembly models do not capture the rapid length equalization after severing observed experimentally. This is shown in the new Figure 3, which focuses on the short-time behavior and neglects protein replenishment. This has allowed us to rule out both cases in which one protein is shared and the other is separate, thereby excluding all of the constant disassembly models.

Reviewer #2:The authors have analyzed a class of models for length control of the two cilia of Chlamydomonas. The interesting result is that a naive model in which both tubulin and motor pools are shared between the two cilia doesn't work: only the summed ciliary length is determined and the individual lengths are undetermined. To get length control for two cilia, the authors add a length-dependent depolymerization, which can be satisfied by a depolymerase whose concentration increases with length (which happens if anterograde transport of the depolymerase is advective but retrograde transport diffusive).My main concern is: what is different/new compared to the Hendel et al. (2018) paper? Is this paper wrong, in the sense that their mechanism (which has no length-dependent depolymerase) does not work, contrary to their claims? If this is the case, then this point must be stressed. If the Hendel et al. paper is correct, then more justification of the novelty of the current work is necessary.

We thank the reviewer for this valuable feedback. We agree that in our original submission we did not effectively distinguish our results from those of earlier papers. We have performed new agent-based simulations to better place our model into the context of the Hendel et al. (2018) model. This has led to a new and surprising result: of the five models discussed in our paper, in fact only one leads to length control consistent with the available experimental data, and it is *not* the one of Hendel et al. Instead, it is the model in which length control is achieved by length-dependent depolymerization (an idea supported by previously published papers on *Chlamydomonas reinhardtii* [2] and *Leishmania major* [3], which both describe the role of a microtubule depolymerizing protein (kinesin-13) in the assembly and length regulation of flagella).

As mentioned above, our revised submission has three main results. First, models in which only one protein is separate do not capture the rapid length equalization observed experimentally, which allows us to rule out all of the constant disassembly models on a case-by-case basis via their short-time behavior.

To capture this absence of length equalization, one must properly account for the asymmetry in protein levels after severing, an effect which was initially missed by us as well as Hendel et al. We have used new agent-based simulations to reproduce the results of Hendel et al. and confirmed that their model involves a control mechanism that rapidly restores the motor pools after severing. In their simulations of the severing experiment, the amount of IFT lost upon cutting is returned immediately to the basal pool. This assumption is required for agreement with the experiments, yet we know of no biological justification for this assumption, which requires the cell to count the number of motors on the flagellum and to maintain this number at a fixed value. This subtlety of simulating the severing experiments, that not only tubulin but also motors must be depleted on severing, was initially missed by us and prevented us from seeing this additional control mechanism.

Our second main result is that protein replenishment is fundamentally inconsistent with constant disassembly, as shown by the simple analytic argument in our previous response. This allows us to rule out constant disassembly models on the basis of the long-time behavior over

which protein replenishment takes place.

It may seem plausible that adding an external control mechanism that replenishes protein levels would lead to length equalization, thus resolving the issue of unequal steady-state lengths after severing. Indeed the final phase of severing experiments implies a replenishment of the pools at some relatively slow timescale *τ_r_*. However, adding such a control mechanism on free proteins levels in the basal body does not lead to length equalization. Instead, in this case the limiting-pool mechanism completely fails to control lengths. This can be shown by the following simple argument: assuming that the cell maintains target protein concentrations in the basal body, in steady-state these concentrations will be equal to their target levels. However, in the constant disassembly model the growth rate is determined solely by basal body concentrations, so that length drops out of the steady-state equations. That is, the constant disassembly is inconsistent not only with the first phase of rapid length equalization, but also with a plausible control mechanism describing the second phase of slow recovery.

More precisely, to include external control with a target tubulin level T¯f and target motor number in the basal body, we include the following equations:τrdTdt=T¯f−TfτrdMdt=M¯f−Mf.

In steady-state, Tf=T¯fand Mf=M¯fso that in steady-state the assembly term γkonM¯fT¯f is constant, and the only way to obtain a balance point is through length-dependent disassembly. Note that, although here we have used a linear restoring force for simplicity, the conclusion also holds for general nonlinear restoring forces having a stable steady-state at Tf=T¯f and Tm=M¯f.

We have included this argument in the revision.

Our third main result is that the active disassembly model we propose is consistent with both the rapid length equalization and the protein replenishment observed in severing experiments; we show that the model agrees with data from Ludington et al. (2012) and Rosenbaum et al. (1969). These three results and the understanding acquired through our new agent-based simulations leads us to believe that all available experimental evidence points in the direction of flagellar length being controlled by length-dependent depolymerization (the last model discussed in our original manuscript). With these new results in hand, we feel that our theoretical work significantly advances our understanding of flagellar length control by challenging the existing paradigm of control by length-dependent polymerization and switching focus to length-dependent depolymerization.

We have included this in the Discussion, and made changes throughout the manuscript to reflect this improved understanding.

The following points need to be addressed in the manuscript1) What is the concentration profile of the depolymerase? Please add this in a figure. I assume that it is a linear increasing concentration from the base (as in the Hendel paper) but I would like to see this.

We have included above the concentration profile of the depolymerase resulting from our new agent-based simulations, which is in agreement to the theoretical prediction. The concentration profile is linear, confirming the reviewer’s comment and the quasi steady-state assumption. This figure has been included in the new Figure 2B(inset) of the revision.

2) Under what conditions can depolymerases regulate a single cilium? Is a limiting tubulin pool necessary? Is a limiting tubulin pool necessary for the case of two cilia? Is a limiting pool of IFT motors necessary? The general point I am raising is that length-dependent depolymerization is a strong assumption and perhaps it is sufficient under very broad conditions. What are they? Then this point needs to be discussed.

As the reviewer points out, unlike the constant disassembly models, in which the limiting-pool mechanism is essential, concentration-dependent disassembly yields length control under mild assumptions including the case that all biomolecules are in excess. However, in severing experiments on *Chlamydomonas* the shortening of the unsevered flagellum shows the importance of depletion effects, and limiting-pools of biomolecules are necessary to capture this effect. This is also demonstrated by experiments of Rosenbaum et al. that use cyclohexamide to block new protein synthesis after severing and result in shorter flagella [4].

Within our model this limiting-pool mechanism may arise in one of three ways: (i) both tubulin and motors are limited, (ii) tubulin is limited and motors are in excess, or (iii) tubulin is in excess and motors are limited. Previously we considered only case (i) in which both biomolecules were in excess. In the new Appendix 3 section “Biomolecules in excess” we show that we may rule out case

(ii) based on experimental data on length-dependent injection rates. The new Appendix 3—figure 2 shows that not only the previously-considered case (i) but also the case (iii) in which only motors are limited is consistent with the severing experiments. Therefore, to in response to the reviewer’s question, we find while in principle depolymerases are sufficient to regulate lengths of two flagella, a limiting pool of IFT proteins is necessary for agreement with data, whereas tubulin may either be limited or in excess.

In addition to studying whether proteins may be in excess in the active disassembly model, we have also investigated how proteins present in limiting amounts may be shared between basal pools. In the new Appendix 3—figure 1, we show that if both tubulin and IFT proteins are present in limiting amounts, the basal pools of both of these biomolecules must be shared in order to obtain the rapid length equalization observed in data.

3) The stochastic aspect of the work is irrelevant as it is not used and should be removed from the paper (perhaps put into another paper).

In the new version, we have emphasized the deterministic aspect of this work by removing the Langevin (SDE) formulation, which we plan to include in a more technical follow-up paper. We have replaced the individual simulated SDE trajectories by the solution of the deterministic ODE, together with the results of our new agent-based stochastic simulations, and removed the fluctuation analysis from the Appendix.

Reviewer #3:The paper describes a very interesting study of the possible models that can account for the simultaneous length control in the flagella of a single-cell organism. The study is illuminating, and shows how simple models can help to shed light on the microscopic processes, simply from the analysis of large-scale dynamics. I have a few comments:

We appreciate the reviewer’s feedback and enthusiastic response. Note that we have made significant changes in response to reviewers #1 and #2 as detailed above.

1) How many MT are in each flagellum? How is this number controlled? Is this also a dynamic variable that self-organizes? The implicit assumption is that this is a constant (determined at the base, and therefore not dependent on length?), but should be explained.

The axoneme (the microtubule structure of eukaryotic cilia and flagella) has 9 outer microtubule doublets surrounded by an inner microtubule central pair (often referred to as the 9+2 structure) [6], the detailed ultrastructure of which having been recently characterized by cryo-EM microscopy [7, 8]. Although there are subtle variations in structure from the base to tip, given this highly regular structure it is reasonable to treat the cross-section as fixed and length-independent. We have included this clarification in the revised manuscript.

2) The authors consider that the ballistic motion of the motors to the growing tip is uninterrupted, is motors and their cargo do not detach from each other or from the MT track until the tip. Is this known to be a good approximation of this system? It certainly simplifies the analysis, as they do not need to solve the spatial distribution of the density of motors, and cargo, along the flagella's length.I would suggest discussing how this is different from models of the growth and steady-state of actin-based protrusions, where dis-assembly is at the base, and the length is a result of force balance (see Naoz et al., 2008; Orly et al., 2014).

The ballistic motion of the motors kinesin-2 and dynein responsible for IFT is uninterrupted and processive, as evidenced by kymographs containing the trajectories of IFT cargo [9, 10]. As a proof of concept, we assume within our model that the depolymerizing protein has the same motion as kinesin-2 –uninterrupted ballistic motion to the tip followed by diffusive motion to the base – resulting in a linear concentration profile. However, the ballistic-to-diffusive assumption is not essential for the model; so long as the depolymerizer has a concentration gradient (e.g. an exponential profile such as the one described in the papers cited by the reviewer), the conclusion of simultaneous length control holds. This is indeed an important point as it highlights the robustness of the model and allows for candidate depolymerizers with different pattern of motion e.g. intermittent rather than uninterrupted ballistic motion from the base to tip. We have highlighted this point and cited the corresponding references in the revision.

3) The question about the return current of tip-directed motors is also an open issue for myosin motors along actin-filled protrusions. Interesting dynamics and traffic jams have been observed and modeled (see Yochelis et al., 2015; Pinkoviezky and Gov, 2014, 2017), I suggest contrasting with the situation in the flagella.

We thank the reviewer for pointing out these references. It is illuminating to see some of the phenomena that arise in analogous actin-myosin systems, and we have cited this previous work in the Discussion.

4) The noise in the reactions that control the growth at the tip should give rise to a term in (1) that has noise multiplied by J and T_f_? Would this change the dynamics?

In our revised manuscript, we have replaced the SDE model with agent-based stochastic simulations that explicitly model the fluxes in motor and tubulin numbers and their associated fluctuations. The full simulations are in good agreement with the deterministic equations and show that the dynamics is not changed significantly by the effects of multiplicative noise.

5) MTs undergo catastrophes and recoveries, and the data traces seem to show this. Does this type of dynamics matter for the model? This should be discussed.

Early studies of axonemes showed that the microtubule doublets in the axoneme are highly stable [11]. More recent work, in which the researchers were able to purify axonemal tubulin, showed that although purified axonemal tubulin has slower dynamics and lower catastrophe frequency, it still undergoes dynamic instability [12]. It is possible that post-translational modifications are responsible for the additional stability, but this is still an area of active research.

[Editors’ note: what now follows is the decision letter after the authors submitted for further consideration.]

Summary:[…]However, the paper also has some serious weaknesses. In particular:1) The experimental basis for their proposal of length dependent depolymerization are much weaker than claimed. In fact, Piao et al., 2009 and 2013, do not give evidence for depolymerases at the flagellar tip in steady state.

Recent experiments in *Giardia* (McInally et al., 2019), which has four pairs of cilia and an IFT machinery similar to *Chlamydomonas*, have shown that kinesin-13 is involved in length regulation via its disassembly activity and is carried by IFT proteins. This observation made us anticipate its role in the disassembly of cilia in *Chlamydomonas* as well. Although the work of McInally was unpublished as of our previous submission, it is now available as a bioRxiv preprint and therefore we are able to cite it freely.

However, we acknowledge that we may have been too quick to suggest kinesin-13 plays a similar role in *Chlamydomonas*. Given the Wang et al. (2013) reference pointed out by the reviewer, which reports only negligible amounts of kinesin-13 in the flagellum at steady-state, any role for kinesin-13 in length control is uncertain. Therefore, we have removed statements identifying kinesin-13 as the depolymerizer throughout the manuscript and instead refer to a general and as of yet unknown depolymerizer. This is discussed in the following paragraph of the revision:

“Further experiments such as the single molecule turnaround experiments pioneered recently in *C. elegans* (Mijalkovic et al., 2018) are needed to establish the identity of the hypothesized depolymerizer. While recent experiments have shown kinesin-13 to be involved in length control in *Giardia* (McInally et al., 2019), the observation that only negligible amounts of flagellar kinesin-13 are present at steady-state (Wang et al., 2013) appears to preclude it from being the candidate depolymerizer of our model.”

2) The claim that length dependent depolymerization is the only mechanism to account for the data is an overstatement.

We thank the reviewer for this important comment. In the revision we have narrowed the scope of our claims by more clearly stating our assumptions and by delimiting the space of models under consideration.

As we now explain in the revision:

“Herein, we limit our theoretical exploration to the space of models defined by the following processes: IFT particle assembly and injection at the flagellar base, motion of IFT proteins along the flagellum, and tubulin polymerization and depolymerization at the flagellar tip (see schematic Figure 2A). We further assume that IFT particle injection satisfies first-order chemical kinetics and allow for a control mechanism that regulates protein levels in the basal pool. Note that this model space does not include all possibilities. In particular, it does not include the time-of-flight model (Wren et al., 2013), in which additional reactions affect protein state inside the flagellum (e.g. proteins enter in an activated state and deactivate at some rate).”

We have also refined our claim in the revision to say that *within this model space*, the models with constant disassembly break down whereas length-dependent depolymerization allows for agreement with data:

“Within the model space outlined, our main results hold independent of these details. Notably, we find that length-independent disassembly of microtubules cannot account for the experimental results, whereas incorporating length-dependent disassembly (e.g. through the ballistic-to-diffusive motion of a depolymerizing protein) leads to reasonable agreement with the experiments.”

3) The discussion of previous work (Hendel et al.) that did find length regulation as seen in experiments without the need of length dependent depolymerization is discussed only superficially and with a somewhat negative attitude. This previous work should be taken more seriously and discussed more carefully. It does show that length regulation does not require length dependent depolymerization. However, this seems to be resulting from different details and assumptions in the model.

The work of Hendel et al. and its predecessors including Marshall et al. (2001) and Marshall et al. (2005) serve as the intellectual foundation of our work. These models have inspired us to spend a considerable amount of time reproducing the results of Hendel et al. and understanding their assumptions, and our discussion and critiques of the model are in the spirit of building on this foundational work.

In the revision, we have elaborated on our discussion of Hendel et al. We have clarified that we have followed the simulation protocol described in their paper and are able to replicate their results using a constant disassembly model with shared tubulin and separate motors provided that the total number of motors are instantaneously replenished to their original amounts upon severing. However, whereas the Hendel et al. model does not yield length equalization in the case of no protein replenishment (see Figure 3Biii), severing experiments show that length equalization occurs even when protein synthesis is blocked using cyclohexamide (Rosenbaum et al., 1969). Note further that controlling motor number in the flagellum is not equivalent to controlling protein concentrations in the basal pool; when we modify the Hendel et al. model by replacing the control on flagellar motor number by a control on motor concentration in the flagellar base, the model breaks down as described in the Results section titled “Controlling protein levels in the basal pool is incompatible with our constant disassembly models”.

The following explanation is included in the revision:

“We remark on the differences between our model and (Hendel et al., 2018), in which simultaneous length control was obtained using a balance point model with constant disassembly. […] As shown in Results, when basal pool concentrations are controlled according to (18)–(19), there is a breakdown of length control for the constant disassembly models.”

We further explain:

“The model equations (57)–(58) [corresponding to shared tubulin pools and separate motor pools with constant disassembly] have a similar form to existing models (Marshall et al., 2005; Hendel et al., 2018), in which the assembly rates involve a factor of *T* − *L*_1_ − *L*_2_ and either a 1/*L_i_*or 1/*L*^2^*_i_*-dependence in the denominator, for *i* = 1, 2 as discussed earlier in the context of a single growing flagellum. Although the equations are similar, the absence of length equalization in our model (Figure 3Biii) contrasts with the length equalization achieved in Hendel et al. by an additional control mechanism that instantaneously replenishes the number of motors on the flagellum after severing. As noted in the Discussion, the importance of protein replenishment for the model appears to be inconsistent with experimental results (Rosenbaum et al., 1969), which show that length equalization occurs even in the absence of new protein synthesis.”

The authors should tone down their claims and more carefully relate their results to previous work. Then this paper could become a very valuable contribution that clarifies subtle but general aspects of length regulation and provides novel insights in the possible role of length dependent depolymerization.The authors have to revise their manuscript carefully before it could be suitable for publication in eLife.Essential revisions:– The authors discuss a mechanism for length dependent depolymerization. However, several arguments are unclear. The authors begin with the assumption in the Introduction: "We will assume for now that the rate-limiting protein is kinesin-2, which is the molecular motor responsible for transport toward the tip of the flagellum". But, then in subsection “Tubulin shared, motors shared and concentration-dependent disassembly” the authors write "although to this point we have assumed the rate-limiting protein is kinesin-2, in fact the model is valid for any rate-limiting IFT protein. Therefore, in what follows we assume that the rate-limiting protein is a depolymerizer having the same motion as kinesin-2, uninterrupted ballistic motion to the tip followed by diffusive motion to the base…"The authors should clarify if there is a family of kinesins that possess both these properties, namely, anterograde transport of tubulin dimers in a directed manner along a microtubule as well as microtubule depolymerase activity. It seems no such motor is known.

We thank the reviewer for pointing out this issue and will now clarify the mechanism of depolymerizer transport we have in mind. We do not assert that a family of kinesins exists that achieves both anterograde transport of tubulin and microtubule depolymerization. Indeed, we do not know of a kinesin possessing both of these properties. Rather, the mechanism we consider to be most likely is that the depolymerizer is a non-motile protein, which is transported ballistically to the flagellar tip as IFT cargo and diffuses back to the flagellar base. In this scenario, the ballistic speed in the anterograde direction is determined by the speed at which kinesin-2 transports IFT particles. (Another possibility, discussed below in response to the reviewer comments but not the main focus of this work, is that the depolymerizer itself is motile). This clarification has been added to the revision, where we state:

“We will assume that the depolymerizer has the same motion as kinesin-2 – uninterrupted ballistic motion to the tip followed by diffusive motion to the base – resulting in a linear concentration profile. This would be the case for any non-motile protein that is transported ballistically to the flagellar tip as IFT cargo and diffuses back to the flagellar base.”

Kinesin-13 is an example of such a non-motile depolymerizer, and it is known to be transported into the flagellum via IFT (Piao, 2009). Also, it has recently been demonstrated to be involved in length control in *Giardia* (McInally et al., 2019). However, as mentioned above, the evidence available for *Chlamydomonas* does not indicate that kinesin-13 is involved in length control. Therefore, although our model suggests that such a non-motile, depolymerizing IFT protein could solve the length control problem, further investigation is needed to determine if such a protein is responsible for length control in *Chlamydomonas*. These qualifications have been added to the revised manuscript, where we state:

“Although our results suggest that having a depolymerizer – one which is ballistically transported to the tip and then diffuses back – provides an appealing model for simultaneous length control, such a depolymeriser has yet to be identified in *Chlamydomonas*.”

We have also clarified our discussion of the rate-limiting protein for IFT and its relationship to the depolymerizer. Previously, we assumed that the rate-limiting protein for IFT was the depolymerizer. However, this assumption does not cause any loss of generality; as we now explain, the same formulas are obtained with rescaled parameters in the more realistic scenario that the rate-limiting protein and the depolymerizer are different. As stated in the revision:

“In the description of the concentration-dependent disassembly model in Results, for convenience we made the assumption that the depolymerizer is the rate-limiting IFT protein. […] Therefore the equations of the manuscript remain valid in this more general case upon making the identification

d0→d0+d1⁢c0′ and d1→K⁢DD′⁢d1.”

We have also verified this result in agent-based simulations that include different populations of depolymerizer and rate-limiting IFT protein, with depolymerizers present in excess in the basal pool, carried to the flagellar tip by IFT, and diffusing back. These simulations result in simultaneous length control, as we now show in the revision in Appendix 3—figure 3A.

– In support of their claim, the authors quote the experimental observations of Piao et al. Piao et al. clearly state in their paper that kinesin-13 "was transported by IFT into flagella during flagellar shortening." Further elaborating on this mechanism, Wang et al. (2013) reported that "CrKin13 was barely detectable in the flagella of steady state cells, as shown previously (Piao et al., 2009). However, during rigorous phase of flagellar assembly at 15 and 30 min after deflagellation, CrKin13 was found to be enriched in the flagella and decreased to normal level in fully assembled flagella". Thus, unlike the mechanism proposed by Fai et al. in this paper, experiments indicate very little presence of kinesin-13 depolymerases in the flagella under normal circumstances. Only amputation of one of these trigger a signal that leads to rapid entry of the kinesin-13 into the uncut flagellum for its depolymerization that supplies tubulins for the initial regeneration of the amputated flagellum. Thus, the roles of kinesin-2 and kinesin-13 and their presence or absence in the flagellum should be treated separately and cannot be represented by an all encompassing single motor. The relation to experiments and the required properties of motors should be discussed more carefully (see also next point)).

As mentioned above, recently published experimental data identified kinesin-13 as a depolymerizer involved in flagellar length control in *Giardia*, whose four pairs of flagella have different steady-state lengths (McInally et al., 2019). When CRISPRi-mediated knockdown was used to deplete kinesin-13 levels by approximately 60%, the flagellar lengths of each of these pairs was found to decrease by 5–20%. This led us to speculate on a similar role for kinesin-13 in *Chlamydomonas*.

However, we acknowledge that the evidence from Piao et al. (2009) and Wang et al. (2013) appears to preclude kinesin-13 from being involved in length control in *Chlamydomonas*. Therefore, as mentioned above, we have removed the previous statements identifying kinesin-13 as the depolymerizer and refer instead to a general and as of yet unidentified depolymerizer.

We have also disentangled the roles of kinesin-2 and the depolymerizer as discussed in detail in the response to the previous comment.

– Piao et al. demonstrated the depolymerase activity of kinesin-13 family (and not kinesin-8 family) in the microtubule disassembly in cilia. Kinesin-13 diffuses along microtubules to target either end and are not known to transport tubulin. Therefore, the claim of experimental support of the model seems to be a too strong statement.

As mentioned above, we have removed the claims of experimental support for any particular depolymerizing protein and generalized our model to the case of a non-motile depolymerizer different from the rate-limiting IFT protein and carried by IFT alongside tubulin.

– Subsection “Tubulin shared, motors shared and concentration-dependent disassembly”, the authors state "Unlike constant disassembly models in which a limiting-pool mechanism is essential for length control,…" This appears to be an incorrect statement. Some constant-disassembly models, e.g., Time-of-flight (TOF) model, which also assumes constant disassembly, does not assume "limiting pool". Instead TOF assumed "differential loading" of IFT particles, as indicated by the experiments of Wren et al.

We thank the reviewer for this comment and have corrected our claims. As mentioned above, we have better delimited the model space under consideration and acknowledged that mechanisms such as time-of-flight fall outside the space of models we consider.

Therefore, the statement "As we shall see, the constant disassembly models do not result in the rapid length equalization observed experimentally" is an overstatement and oversimplification.

We have qualified this claim by restricting it to the model space we have explored. We have edited this statement, which now reads:

“As we shall see, the constant disassembly models within our model space do not result in the rapid length equalization observed experimentally.”

– The key assumptions of Fai et al.'s model seem to be the following: (a) a concentration gradient of the depolymerases along the length of the flagellum and (b) the local depolymerization rate is proportional to the local concentration of the depolymerases. There appears to be an additional implicit assumption: the depolymerase is non-processive in its depolymerase activity (not to be confused with processivity in motility). Otherwise, depolymerases loaded on to the MT plus-end at a location closer to the ciliary tip may continue to depolymerize even when the MT becomes shorter and its plus-end reaches points where the depolymerase concentrations are quite different.

The reviewer is correct that the assumption that depolymerase activity depends on local concentration excludes processive depolymerizers that could remove more than a few tubulin subunits from the flagellum before falling off into a deactivated state. To investigate whether this is a fundamental restriction, we have used our agent-based model to explore a mechanism of processive depolymerization, in which depolymerizers within 1 micron of the flagellar tip may bind and begin depolymerizing at a fixed rate. The depolymerization duration is randomly drawn from an exponential distribution with prescribed mean. As we now state in the revision and show in the Appendix 3—figure 3C:

“In Appendix 3—figure 3C we show the results of simulations using mean depolymerization times from 10 s (corresponding to 0.01 µm) up to 10 min (corresponding to 0.6 µm). For mean depolymerization times up to 100 s, we find the results to be qualitatively similar to the nonprocessive case, whereas for longer mean depolymerization times the steady-state lengths exhibit oscillations about the steady-state. The emergence of oscillations is not entirely surprising since processive depolymerization effectively introduces into the equations a time delay, which is known to give rise to oscillations in many contexts (Richard, 2003).”

Further, the model is not fundamentally restricted to a proportional dependence of depolymerase activity on concentration. If instead the local depolymerization rate were a non-linear function of concentration, a Taylor series expansion may be performed as in (Klein et al., 2005) to obtain the corresponding linearized system discussed in the Appendix. We have clarified these points in the revision, where we say:

“Although here we have taken depolymerase activity to depend linearly on concentration, the model generalizes to the non-linear case in a straightforward manner. For a local depolymerization rate that is an arbitrary function of concentration, a Taylor series expansion may be performed as in (Klein et al., 2005) to obtain the corresponding linearized system discussed in Appendix 3.”

– The possibility of a concentration gradient of the motors arose naturally already in the paper of Hendel et al. (see page 667 of Hendel et al.). However, Hendel et al., did not associate this concentration gradient with that of depolymerases, which seems to be the appropriate picture.

We thank the reviewer for this comment. To make this point clear, we have included the following in the revision:

“How proteins can develop and maintain such concentration profiles is therefore one of the key questions raised by the model. Indeed, concentration gradients (unassociated with depolymerizing activity) were observed already in (Hendel et al., 2018) as the result of ballistic-to-diffusive motion.”

Note that the concentration-dependent disassembly is not limited to the case of a linear concentration distribution, as we now explain below and demonstrate in the new Appendix 3—figure 3B:

“Although here as proof of principle this concentration gradient is achieved by the mechanism of ballistic-to-diffusive motion, our main results are independent of the detailed form of this concentration gradient and how it is generated. As shown in the new Appendix 3—figure 3B, our model allows for depolymerizers with nonlinear concentration profiles and different patterns of motion, e.g. exponential concentration distributions such as those recently observed in *Giardia* (McInally et al., 2019) and those generated by motile proteins that bind and unbind to cytoskeletal filaments, as theorized in the context of actin-myosin systems (Naoz et al., 2008; Orly et al., 2014; Pinkoviezky and Gov, 2014;, Pinkoviesky and Gov, 2017; Yochelis and Gov, 2017).”

We have specified the form of this exponential dependence in the following section added to the Appendix: “To show that our model allows for non-linear concentration gradients, we set up an agent based simulations which would lead to an exponential concentration distribution, similar to (Naoz et al., 2008). […] The results are found to be qualitatively similar to the case of a linear concentration gradient, supporting our claim that concentration-dependent disassembly is able to control lengths independent of the precise form of the concentration gradient.”

– The authors present "testable predictions". However there are some problems. The experimental results already reported by Piao et al., 2009 and 2013, do not provide evidence for the presence of depolymerases at the tip in steady state. Also based on these experiments the existence of a concentration gradient of the depolymerases in the steady-state (that too, an increasing concentration towards the tip) seems to be unlikely.The second testable prediction that the authors mention would not be a clear test for validation/refutation of their model. For all those models that depend on a dynamic balance of the rates of polymerization and depolymerization in the steady-state of the flagellum, lowering the entry flux of the IFT particles would lead to overall depolymerization although the local rate of depolymerization of the microtubules at the plus end would remain unchanged.

As mentioned above, we have replaced the references to kinesin-13 by a more general and as of yet unidentified depolymerizer. We describe experiments that may be helpful for searching for likely candidates. In the revision we now state:

“In addition to our claim that the depolymerization rate is non-constant and dependent on length, our model leads to testable predictions that may be useful in identifying candidate depolymerizers. For example, according to our model the depolymerizer active in length control is not uniformly distributed along the flagellum; its concentration increases toward the flagellar tip. This could be tested experimentally by fluorescently labeling candidate depolymerizers and studying their concentration profiles along the flagellum as recently done to characterize the concentration profile of kinesin-13 in *Giardia* (McInally et al., 2019).”

Further, we have clarified that:

“Our model predicts that the tubulin and depolymerizer pools must both be shared for the concentration-dependent disassembly model to capture the rapid length equalization observed (Appendix 3). This illustrates the dramatic consequences in behavior that can occur when biomolecules are shared between compartments, and highlights the importance of knowing which proteins are exchanged between the basal pools in the context of flagellar length control. In particular, having a protein that is *not* exchanged can provide simultaneous length control by a limiting-pool mechanism, but it introduces asymmetries that contradict the length equalization observed in severing experiments.”

– The slight difference in the expressions for L_ss_ derived by Hendel et al. and that of Fai et al. arises from difference in the scenarios considered by the two. The two assumptions made by Hendel et al. were clearly stated in their paper. The first assumption of "a constant source of free motor protein at the tip" and the second assumption of "the approximation in which motors that have reached the base immediately transport back to the tip" are exactly the two conditions ("no tubulin depletion" and "instantaneous ballistic motion") that Fai et al. point in reducing their result to that of Hendel et al. In this sense their derivation is only slightly improved compared to that of Hendel et al.

In fact, there are subtle differences between our expressions and those of Hendel et al., e.g. the constant term in the denominator that prevents the flux from becoming singular at *L* = 0 even in the instantaneous ballistic motion limit *v* → ∞. However, we agree that these differences are not the main results of our work. As discussed above, the key difference between Hendel et al. and our constant disassembly model with shared tubulin and separate motor pools is that whereas they effectively control the number of motors loaded on the flagellum, we only allow for control on basal protein levels (which we show is incompatible with their model).

We agree that the previous wording was confusing, as it overly emphasized the subtle differences in these formulas. For clarification, we have now added the following sentence after the discussion of the different formulas obtained by us and Hendel et al.:

“Note however that the essential difference between our model and Hendel et al. (2018) lies in their effective control mechanism on the number of motors loaded on the flagella, which is not captured by any differences in these formulas (see Discussion).”

– I find the claims of the paper that Hendel’s model is somehow incorrect and that the present manuscript solves all open issues unconvincing. Hendel et al. show a scenario where length control works, probably based on different detailed assumptions. This should be more clearly discussed and not superficially mentioned as in the paragraph three of the Discussion.

As mentioned above, we have more clearly delineated why the control mechanism used by Hendel et al. to maintain a constant number of motors loaded on the flagellum falls outside our scope of models, which only allows for control mechanisms on protein levels in basal pools. Moreover, we have shown that seemingly analogous models that would fall within our model space, e.g. those that control basal protein concentrations, or those in which all biomolecules are shared, do not achieve simultaneous length control under the constant disassembly assumption.

– The authors discuss the case where molecular components stabilize their levels (Equations 18 and 19) and point out the problem that in steady state such a relaxation to fixed levels breaks length regulation. It remains unclear if that is also a problem for the proposed model based on length dependent depolymerization. Also, it would be important to know what happens when the motors alone are relaxing to a fixed level but the tubulin is limited and fixed. Would that be similar to the results of Hendel et al.?

Unlike the constant disassembly models we have considered, the depolymerizer model leads to a unique steady-state length in the presence of protein replenishment. This is demonstrated in Figure 4C and in Figure 4—video 2, as we now state in the revision:

“Another feature of the concentration-dependent disassembly model is that it allows for an external control mechanism on protein levels in the basal pool, unlike the constant disassembly models we have considered. As shown in Figure 4C and Figure 4—video 2, upon including protein replenishment via eq. (18)–(19) on a timescale of *τ_r_*= 10 mins, the recovery of the flagella back to their original lengths is in reasonable agreement with experimental data.”

Note that in the absence of protein replenishment, the depolymerizer model yields rapid length equalization (Figure 4B) whereas the constant disassembly model comparable to Hendel et al. does not (Figure 3Biii).

The case of motors replenishing to a fixed level but tubulin limited and fixed does not reduce to the model of Hendel et al.: in their simulations, Hendel et al. replenish tubulin to restore the flagellar lengths back to their initial values.

– The authors seem to miss relevant references: Varga et al. (2009); Klein et al. (2005); Johann et al. (2012).‏ These should be cited and their consequences for this paper should be discussed.

We thank the reviewers for pointing out these references. We have discussed their relevance to the paper as follows in the revision:

“Although here we have taken depolymerase activity to depend linearly on concentration, the model generalizes to the non-linear case in a straightforward manner. […] The aggregation of motile depolymerizing proteins has also been demonstrated in previous theoretical studies of microtubule length control (Klein et al., 2005; Johann et al., 2012; Reese et al., 2014); note however that these previous works differed from our model of flagellar IFT in that they considered isolated microtubules surrounded by a constant concentration bath and/or significant steric interactions between motile proteins.”

**References**

[1] W. F. Marshall, H. Qin, M. R. Brenni, J. L. Rosenbaum, Flagellar length control system: testing a simple model based on intraflagellar transport and turnover, Molecular Biology of the Cell 16 (1) (2005) 270–278.

[2] T. Piao, M. Luo, L. Wang, Y. Guo, P. Li, W. J. Snell, J. Pan, et al., A microtubule depolymerizing kinesin functions during both flagellar disassembly and flagellar assembly in *Chlamydomonas*, Proceedings of the National Academy of Sciences 106 (12) (2009) 4713–4718.

[3] C. Blaineau, M. Tessier, P. Dubessay, L. Tasse, L. Crobu, M. Pages, P. Bastien, A novel` microtubule-depolymerizing kinesin involved in length control of a eukaryotic flagellum, Current Biology 17 (9) (2007) 778–782.

[4] J. L. Rosenbaum, J. E. Moulder, D. L. Ringo, Flagellar elongation and shortening in *Chlamydomonas*: the use of cycloheximide and colchicine to study the synthesis and assembly of flagellar proteins, The Journal of Cell Biology 41 (2) (1969) 600–619.

[5] W. B. Ludington, L. Z. Shi, Q. Zhu, M. W. Berns, W. F. Marshall, Organelle size equalization by a constitutive process, Current Biology 22 (22) (2012) 2173–2179.

[6] D. W. Fawcett, K. R. Porter, A study of the fine structure of ciliated epithelia, Journal of Morphology 94 (2) (1954) 221–281.

[7] K. H. Bui, H. Sakakibara, T. Movassagh, K. Oiwa, T. Ishikawa, Molecular architecture of inner dynein arms in situ in *Chlamydomonas reinhardtii* flagella, The Journal of Cell Biology 183 (5) (2008) 923–932.

[8] C. F. Barber, T. Heuser, B. I. Carbajal-Gonzalez, V. V. Botchkarev Jr, D. Nicastro, Three-´ dimensional structure of the radial spokes reveals heterogeneity and interactions with dyneins in *Chlamydomonas* flagella, Molecular Biology of the Cell 23 (1) (2012) 111–120.

[9] W. Dentler, Intraflagellar transport (ift) during assembly and disassembly of *Chlamydomonas* flagella, The Journal of Cell Biology 170 (4) (2005) 649–659.

[10] W. B. Ludington, K. A. Wemmer, K. F. Lechtreck, G. B. Witman, W. F. Marshall, Avalanche-like behavior in ciliary import, Proceedings of the National Academy of Sciences (2013) 201217354.

[11] O. Behnke, A. Forer, Evidence for four classes of microtubules in individual cells, Journal of Cell Science 2 (2) (1967) 169–192.

[12] R. Orbach, J. Howard, The dynamic and structural properties of axonemal tubulins support the high length stability of cilia, bioRxiv (2018) 451351.